# Adversarially Robust Generalization Requires More Data

**Ludwig Schmidt**
UC Berkeley
ludwig@berkeley.edu

**Shibani Santurkar**
MIT
shibani@mit.edu

**Dimitris Tsipras**
MIT
tsipras@mit.edu

**Kunal Talwar**
Google Brain
kunal@google.com

**Aleksander Mądry**
MIT
madry@mit.edu

## Abstract

Machine learning models are often susceptible to adversarial perturbations of their inputs. Even small perturbations can cause state-of-the-art classifiers with high "standard" accuracy to produce an incorrect prediction with high confidence. To better understand this phenomenon, we study adversarially robust learning from the viewpoint of generalization. We show that already in a simple natural data model, the sample complexity of robust learning can be significantly larger than that of "standard" learning. This gap is information theoretic and holds irrespective of the training algorithm or the model family. We complement our theoretical results with experiments on popular image classification datasets and show that a similar gap exists here as well. We postulate that the difficulty of training robust classifiers stems, at least partially, from this inherently larger sample complexity.

## 1 Introduction

Modern machine learning models achieve high accuracy on a broad range of datasets, yet can easily be misled by small perturbations of their input. While such perturbations are often simple noise to a human or even imperceptible, they cause state-of-the-art models to misclassify their input with high confidence. This phenomenon has first been studied in the context of secure machine learning for spam filters and malware classification [7, 16, 35]. More recently, researchers have demonstrated the phenomenon under the name of *adversarial examples* in image classification [21, 51], question answering [28], voice recognition [12, 13, 49, 62], and other domains (for instance, see [2, 4, 14, 22, 25, 26, 32, 60]). Overall, the existence of such adversarial examples raises concerns about the robustness of current classifiers. As we increasingly deploy machine learning systems in safety- and security-critical environments, it is crucial to understand the robustness properties of our models in more detail.

A growing body of work is exploring this robustness question from the security perspective by proposing *attacks* (methods for crafting adversarial examples) and *defenses* (methods for making classifiers robust to such perturbations). Often, the focus is on deep neural networks, e.g., see [11, 24, 36, 37, 41, 47, 53, 59]. While there has been success with robust classifiers on simple datasets [31, 36, 44, 48], more complicated datasets still exhibit a large gap between "standard" and robust accuracy [3, 11]. An implicit assumption underlying most of this work is that the same training dataset that enables good standard accuracy also suffices to train a robust model. However, it is unclear if this assumption is valid.

So far, the *generalization* aspects of adversarially robust classification have not been thoroughly investigated. Since adversarial robustness is a learning problem, the statistical perspective is of integral importance. A key observation is that adversarial examples are not at odds with the standard notion of generalization as long as they occupy only a small total measure under the data distribution. So to achieve adversarial robustness, a classifier must generalize in a stronger sense. We currently do not have a good understanding of how such a stronger notion of generalization compares to standard "benign" generalization, i.e., without an adversary.

In this work, we address this gap and explore the statistical foundations of adversarially robust generalization. We focus on sample complexity as a natural starting point since it underlies the core question of when it is possible to learn an adversarially robust classifier. Concretely, we pose the following question:

> *How does the sample complexity of standard generalization compare to that of adversarially robust generalization?*

Put differently, we ask if a dataset that allows for learning a good classifier also suffices for learning a robust one. To study this question, we analyze robust generalization in two distributional models. By focusing on specific distributions, we can establish information-theoretic lower bounds and describe the exact sample complexity requirements for generalization. We find that even for a simple data distribution such as a mixture of two class-conditional Gaussians, the sample complexity of robust generalization is significantly larger than that of standard generalization. Our lower bound holds for *any* model and learning algorithm. Hence no amount of algorithmic ingenuity is able to overcome this limitation.

In spite of this negative result, simple datasets such as MNIST have recently seen significant progress in terms of adversarial robustness [31, 36, 44, 48]. The most robust models achieve accuracy around 90% against large $\ell_\infty$-perturbations. To better understand this discrepancy with our first theoretical result, we also study a second distributional model with binary features. This binary data model has the same standard generalization behavior as the previous Gaussian model. Moreover, it also suffers from a significantly increased sample complexity whenever one employs *linear* classifiers to achieve adversarially robust generalization. Nevertheless, a slightly non-linear classifier that utilizes thresholding turns out to recover the smaller sample complexity of standard generalization. Since MNIST is a mostly binary dataset, our result provides evidence that $\ell_\infty$-robustness on MNIST is significantly easier than on other datasets. Moreover, our results show that distributions with similar sample complexity for standard generalization can still exhibit considerably different sample complexity for robust generalization.

To complement our theoretical results, we conduct a range of experiments on MNIST, CIFAR10, and SVHN. By subsampling the datasets at various rates, we study the impact of sample size on adversarial robustness. When plotted as a function of training set size, our results show that the standard accuracy on SVHN indeed plateaus well before the adversarial accuracy reaches its maximum. On MNIST, explicitly adding thresholding to the model during training significantly reduces the sample complexity, similar to our upper bound in the binary data model. On CIFAR10, the situation is more nuanced because there are no known approaches that achieve more than 50% accuracy even against a mild adversary. But as we show below, there is clear evidence for overfitting in the current state-of-the-art methods.

Overall, our results suggest that current approaches may be unable to attain higher adversarial accuracy on datasets such as CIFAR10 for a fundamental reason: the dataset may not be large enough to train a standard convolutional network robustly. Moreover, our lower bounds illustrate that the existence of adversarial examples should not necessarily be seen as a shortcoming of specific classification methods. Already in a simple data model, adversarial examples *provably* occur for any learning approach, even when the classifier already achieves high standard accuracy. So while vulnerability to adversarial $\ell_\infty$-perturbations might seem counter-intuitive at first, in some regimes it is an unavoidable consequence of working in a statistical setting.

## 1.1   A motivating example: Overfitting on CIFAR10

Before we describe our main results, we briefly highlight the importance of generalization for adversarial robustness via two experiments on MNIST and CIFAR10. In both cases, our goal is to learn a classifier that achieves good test accuracy even under $\ell_\infty$-bounded perturbations. We follow

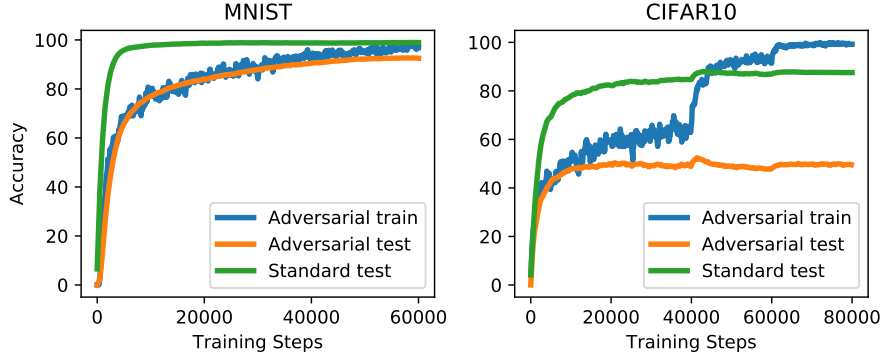

Figure 1: Classification accuracies for robust optimization on MNIST and CIFAR10. In both cases, we trained standard convolutional networks to be robust to $\ell_\infty$-perturbations of the input. On MNIST, the robust test error closely tracks the corresponding training error and the model achieves high robust accuracy. On CIFAR10, the model still achieves a good natural (non-adversarial) test error, but there is a significant generalization gap for the robust accuracy. This phenomenon motivates our study of adversarially robust generalization.

the standard robust optimization approach [6, 36, 54] – also known as adversarial training [21, 51] – and (approximately) solve the saddle point problem

$$\min_{\theta} \mathbb{E}_{x} \left[ \max_{\|x'-x\|_\infty \leq \varepsilon} \mathrm{loss}(\theta, x') \right]$$

via stochastic gradient descent over the model parameters $\theta$. We utilize projected gradient descent for the inner maximization problem over allowed perturbations of magnitude $\varepsilon$ (see [36] for details). Figure 1 displays the training curves for three quantities: (i) adversarial training error, (ii) adversarial test error, and (iii) standard test error.

The results show that on MNIST, robust optimization is able to learn a model with around 90% adversarial accuracy and a relatively small gap between training and test error. However, CIFAR10 offers a different picture. Here, the model (a wide residual network [61]) is still able to fully fit the training set even against an adversary, but the generalization gap is significantly larger. The model only achieves 47% adversarial test accuracy, which is about 50% lower than its training accuracy.[1] Moreover, the standard test accuracy is about 87%, so the failure of generalization indeed primarily occurs in the context of adversarial robustness. This failure may be surprising particularly since properly tuned convolutional networks rarely overfit much on standard vision datasets.

## 1.2 Outline of the paper

In the next section, we describe our main theoretical results at a high level. Section 3 complements these results with experiments. We discuss related works in Section 4 and conclude in Section 5. Due to space constraints, a longer discussion of related work, several open questions, and all proofs are deferred to the appendix in the supplementary material.

## 2 Theoretical Results

Our theoretical results concern statistical aspects of adversarially robust classification. In order to understand how properties of data affect the number of samples needed for robust generalization, we study two concrete distributional models.

While our two data models are clearly much simpler than the image datasets currently being used in the experimental work on $\ell_\infty$-robustness, we believe that the simplicity of our models is a strength in this context. The fact that we can establish a separation between standard and robust generalization already in our Gaussian data model is evidence that the existence of adversarial examples for neural

networks should not come as a surprise. The same phenomenon (i.e., classifiers with just enough samples for high standard accuracy *necessarily* being vulnerable to $\ell_\infty$- attacks) already occurs in much simpler settings such as a mixture of two Gaussians. Note that more complicated distributional setups that can "simulate" the Gaussian model directly inherit our lower bounds.

In addition, conclusions from our simple models also transfer to real datasets. As we describe in the subsection on the Bernoulli model, the benefits of the thresholding layer predicted by our theoretical analysis do indeed appear in experiments on MNIST as well. Since multiple defenses against adversarial examples have been primarily evaluated on MNIST [31, 44, 48], it is important to note that $\ell_\infty$-robustness on MNIST is a particularly easy case: adding a simple thresholding layer directly yields nearly state-of-the-art robustness against moderately strong adversaries ($\varepsilon = 0.1$), without any further changes to the model architecture or training algorithm.

## 2.1 The Gaussian model

Our first data model is a mixture of two spherical Gaussians with one component per class.

**Definition 1** (Gaussian model)**.** *Let $\theta^\star \in \mathbb{R}^d$ be the per-class mean vector and let $\sigma > 0$ be the variance parameter. Then the $(\theta^\star, \sigma)$-Gaussian model is defined by the following distribution over $(x, y) \in \mathbb{R}^d \times \{\pm 1\}$: First, draw a label $y \in \{\pm 1\}$ uniformly at random. Then sample the data point $x \in \mathbb{R}^d$ from $\mathcal{N}(y \cdot \theta^\star, \sigma^2 I)$.*

While not explicitly specified in the definition, we will use the Gaussian model in the regime where the norm of the vector $\theta^\star$ is approximately $\sqrt{d}$. Hence the main free parameter for controlling the difficulty of the classification task is the variance $\sigma^2$, which controls the amount of overlap between the two classes.

To contrast the notions of "standard" and "robust" generalization, we briefly recap a standard definition of classification error.

**Definition 2** (Classification error)**.** *Let $\mathcal{P} : \mathbb{R}^d \times \{\pm 1\} \to \mathbb{R}$ be a distribution. Then the classification error $\beta$ of a classifier $f : \mathbb{R}^d \to \{\pm 1\}$ is defined as $\beta = \mathbb{P}_{(x,y) \sim \mathcal{P}}[f(x) \neq y]$.*

Next, we define our main quantity of interest, which is an adversarially robust counterpart of the above classification error. Instead of counting misclassifications under the data distribution, we allow a bounded worst-case perturbation before passing the perturbed sample to the classifier.

**Definition 3** (Robust classification error)**.** *Let $\mathcal{P} : \mathbb{R}^d \times \{\pm 1\} \to \mathbb{R}$ be a distribution and let $\mathcal{B} : \mathbb{R}^d \to \mathscr{P}(\mathbb{R}^d)$ be a perturbation set.[2] Then the $\mathcal{B}$-robust classification error $\beta$ of a classifier $f : \mathbb{R}^d \to \{\pm 1\}$ is defined as $\beta = \mathbb{P}_{(x,y) \sim \mathcal{P}}[\exists x' \in \mathcal{B}(x) : f(x') \neq y]$.*

Since $\ell_\infty$-perturbations have recently received a significant amount of attention, we focus on robustness to $\ell_\infty$-bounded adversaries in our work. For this purpose, we define the perturbation set $\mathcal{B}_\infty^\varepsilon(x) = \{x' \in \mathbb{R}^d \,|\, \|x' - x\|_\infty \leq \varepsilon\}$. To simplify notation, we refer to robustness with respect to this set also as $\ell_\infty^\varepsilon$-robustness. As we remark in the discussion section, understanding generalization for other measures of robustness ($\ell_2$, rotations, etc.) is an important direction for future work.

**Standard generalization.** The Gaussian model has one parameter for controlling the difficulty of learning a good classifier. In order to simplify the following bounds, we study a regime where it is possible to achieve good *standard* classification error from a single sample.[3] As we will see later, this also allows us to calibrate our two data models to have comparable standard sample complexity.

Concretely, we prove the following theorem, which is a direct consequence of Gaussian concentration. Note that in this theorem we use a *linear classifier*: for a vector $w$, the linear classifier $f_w : \mathbb{R}^d \to \{\pm 1\}$ is defined as $f_w(x) = \text{sgn}(\langle w, x \rangle)$.

**Theorem 4.** *Let $(x, y)$ be drawn from a $(\theta^\star, \sigma)$-Gaussian model with $\|\theta^\star\|_2 = \sqrt{d}$ and $\sigma \leq c \cdot d^{1/4}$ where $c$ is a universal constant. Let $\widehat{w} \in \mathbb{R}^d$ be the vector $\widehat{w} = y \cdot x$. Then with high probability, the linear classifier $f_{\widehat{w}}$ has classification error at most 1%.*

To minimize the number of parameters in our bounds, we have set the error probability to 1%. By tuning the model parameters appropriately, it is possible to achieve a vanishingly small error probability from a single sample (see Corollary 19 in Appendix D.1).

**Robust generalization.**   As we just demonstrated, we can easily achieve *standard* generalization from only a single sample in our Gaussian model. We now show that achieving a low $\ell_\infty$-*robust* classification error requires significantly more samples. To this end, we begin with a natural strengthening of Theorem 4 and prove that the (class-weighted) sample mean can also be a robust classifier (given sufficient data).

**Theorem 5.** *Let* $(x_1, y_1), \ldots, (x_n, y_n)$ *be drawn i.i.d. from a* $(\theta^\star, \sigma)$-*Gaussian model with* $\|\theta^\star\|_2 = \sqrt{d}$ *and* $\sigma \le c_1 d^{1/4}$. *Let* $\widehat{w} \in \mathbb{R}^d$ *be the weighted mean vector* $\widehat{w} = \frac{1}{n} \sum_{i=1}^n y_i x_i$. *Then with high probability, the linear classifier* $f_{\widehat{w}}$ *has* $\ell_\infty^\varepsilon$-*robust classification error at most 1% if*

$$n \ge \begin{cases} 1 & \text{for} \quad \varepsilon \le \frac{1}{4} d^{-1/4} \\ c_2 \, \varepsilon^2 \sqrt{d} & \text{for} \quad \frac{1}{4} d^{-1/4} \le \varepsilon \le \frac{1}{4} \end{cases} .$$

We refer the reader to Corollary 22 in Appendix D.1 for the details. As before, $c_1$ and $c_2$ are two universal constants. Overall, the theorem shows that it is possible to learn an $\ell_\infty^\varepsilon$-robust classifier in the Gaussian model as long as $\varepsilon$ is bounded by a small constant and we have a large number of samples.

Next, we show that this significantly increased sample complexity is necessary. Our main theorem establishes a lower bound for *all* learning algorithms, which we formalize as functions from data samples to binary classifiers. In particular, the lower bound applies not only to learning linear classifiers.

**Theorem 6.** *Let* $g_n$ *be any learning algorithm, i.e., a function from* $n$ *samples to a binary classifier* $f_n$. *Moreover, let* $\sigma = c_1 \cdot d^{1/4}$, *let* $\varepsilon \ge 0$, *and let* $\theta \in \mathbb{R}^d$ *be drawn from* $\mathcal{N}(0, I)$. *We also draw* $n$ *samples from the* $(\theta, \sigma)$-*Gaussian model. Then the expected* $\ell_\infty^\varepsilon$-*robust classification error of* $f_n$ *is at least* $(1 - 1/d)\frac{1}{2}$ *if*

$$n \le c_2 \frac{\varepsilon^2 \sqrt{d}}{\log d} .$$

The proof of the theorem can be found in Corollary 23 (Appendix D.2). It is worth noting that the classification error $1/2$ in the lower bound is tight. A classifier that always outputs a fixed prediction trivially achieves perfect robustness on one of the two classes and hence robust accuracy $1/2$.

Comparing Theorems 5 and 6, we see that the sample complexity $n$ required for robust generalization is bounded as

$$\frac{c\varepsilon^2 \sqrt{d}}{\log d} \le n \le c'\varepsilon^2\sqrt{d} .$$

Hence the lower bound is nearly tight in our regime of interest. When the perturbation has constant $\ell_\infty$-norm, the sample complexity of robust generalization is larger than that of standard generalization by $\sqrt{d}$, i.e., *polynomial* in the problem dimension. This shows that for high-dimensional problems, adversarial robustness can provably require a significantly larger number of samples.

Finally, we remark that our lower bound applies also to a more restricted adversary. Our proof uses only a single adversarial perturbation per class. As a result, the lower bound provides *transferable* adversarial examples and applies to worst-case distribution shifts without a classifier-adaptive adversary. We refer the reader to Section 5 for a more detailed discussion.

## 2.2   The Bernoulli model

As mentioned in the introduction, simpler datasets such as MNIST have recently seen significant progress in terms of $\ell_\infty$-robustness. We now investigate a possible mechanism underlying these advances. To this end, we study a second distributional model that highlights how the data distribution can significantly affect the achievable robustness. The second data model is defined on the hypercube $\{\pm1\}^d$, and the two classes are represented by opposite vertices of that hypercube. When sampling a datapoint for a given class, we flip each bit of the corresponding class vertex with a certain probability. This data model is inspired by the MNIST dataset because MNIST images are close to binary (many pixels are almost fully black or white).

**Definition 7** (Bernoulli model). *Let $\theta^\star \in \{\pm 1\}^d$ be the per-class mean vector and let $\tau > 0$ be the class bias parameter. Then the $(\theta^\star, \tau)$-Bernoulli model is defined by the following distribution over $(x, y) \in \{\pm 1\}^d \times \{\pm 1\}$: First, draw a label $y \in \{\pm 1\}$ uniformly at random from its domain. Then sample the data point $x \in \{\pm 1\}^d$ by sampling each coordinate $x_i$ from the distribution*

$$x_i = \begin{cases} y \cdot \theta_i^\star & \text{with probability } 1/2 + \tau \\ -y \cdot \theta_i^\star & \text{with probability } 1/2 - \tau \end{cases}.$$

As in the previous subsection, the model has one parameter for controlling the difficulty of learning. A small value of $\tau$ makes the samples less correlated with their respective class vectors and hence leads to a harder classification problem. Note that both the Gaussian and the Bernoulli model are defined by simple sub-Gaussian distributions. Nevertheless, we will see that they differ significantly in terms of robust sample complexity.

**Standard generalization.**    As in the Gaussian model, we first calibrate the distribution so that we can learn a classifier with good *standard* accuracy from a single sample.[4] The following theorem is a direct consequence of the fact that bounded random variables exhibit sub-Gaussian concentration.

**Theorem 8.** *Let $(x, y)$ be drawn from a $(\theta^\star, \tau)$-Bernoulli model with $\tau \geq c \cdot d^{-1/4}$ where $c$ is a universal constant. Let $\widehat{w} \in \mathbb{R}^d$ be the vector $\widehat{w} = y \cdot x$. Then with high probability, the linear classifier $f_{\widehat{w}}$ has classification error at most 1%.*

To simplify the bound, we have set the error probability to be 1% as in the Gaussian model. We refer the reader to Corollary 28 in Appendix F.1 for the proof.

**Robust generalization.**    Next, we investigate the sample complexity of robust generalization in our Bernoulli model. For *linear* classifiers, a small robust classification error again requires a large number of samples:

**Theorem 9.** *Let $g_n$ be a linear classifier learning algorithm, i.e., a function from $n$ samples to a linear classifier $f_n$. Suppose that we choose $\theta^\star$ uniformly at random from $\{\pm 1\}^d$ and draw $n$ samples from the $(\theta^\star, \tau)$-Bernoulli model with $\tau = c_1 \cdot d^{-1/4}$. Moreover, let $\varepsilon < 3\tau$ and $0 < \gamma < 1/2$. Then the expected $\ell_\infty^\varepsilon$-robust classification error of $f_n$ is at least $\frac{1}{2} - \gamma$ if*

$$n \leq c_2 \frac{\varepsilon^2 \gamma^2 d}{\log d/\gamma}.$$

We defer the proof to Appendix F.2. At first, the lower bound for linear classifiers might suggest that $\ell_\infty$-robustness requires an inherently larger sample complexity here as well. However, in contrast to the Gaussian model, non-linear classifiers can achieve a significantly improved robustness. In particular, consider the following thresholding operation $T : \mathbb{R}^d \to \mathbb{R}^d$ which is defined element-wise as

$$T(x)_i = \begin{cases} +1 & \text{if } x_i \geq 0 \\ -1 & \text{otherwise} \end{cases}.$$

It is easy to see that for $\varepsilon < 1$, the thresholding operator undoes the action of any $\ell_\infty$-bounded adversary, i.e., we have $T(\mathcal{B}_\infty^\varepsilon(x)) = \{x\}$ for any $x \in \{\pm 1\}^d$. Hence we can combine the thresholding operator with the classifier learned from a single sample to get the following upper bound.

**Theorem 10.** *Let $(x, y)$ be drawn from a $(\theta^\star, \tau)$-Bernoulli model with $\tau \geq c \cdot d^{-1/4}$ where $c$ is a universal constant. Let $\widehat{w} \in \mathbb{R}^d$ be the vector $\widehat{w} = yx$. Then with high probability, the classifier $f_{\widehat{w}} \circ T$ has $\ell_\infty^\varepsilon$-robust classification error at most 1% for any $\varepsilon < 1$.*

This theorem shows a stark contrast to the Gaussian case. Although both models have similar sample complexity for standard generalization, there is a $\sqrt{d}$ gap between the $\ell_\infty$-robust sample complexity for the Bernoulli and Gaussian models. This discrepancy provides evidence that robust generalization requires a more nuanced understanding of the data distribution than standard generalization.

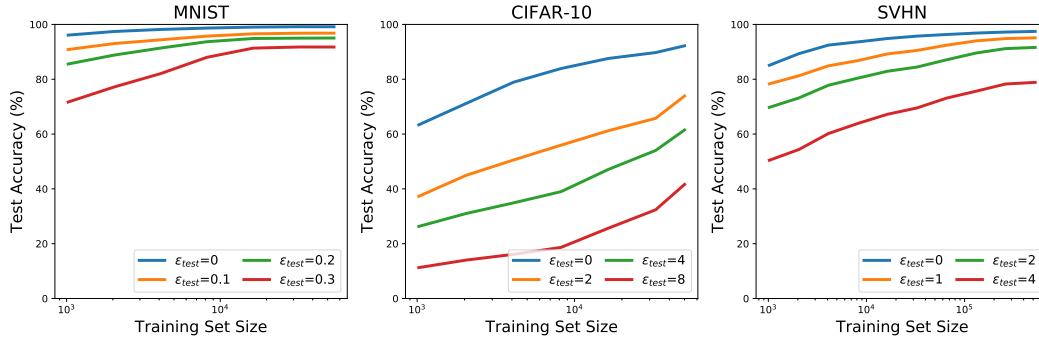

Figure 2: Adversarially robust generalization performance as a function of training data size for $\ell_\infty$ adversaries on the MNIST, CIFAR-10 and SVHN datasets. For each choice of training set size and $\varepsilon_{test}$, we plot the best performance achieved over $\varepsilon_{train}$ and network capacity. This clearly shows that achieving a certain level of adversarially robust generalization requires significantly more samples than achieving the same level of standard generalization.

In isolation, the thresholding step might seem specific to the Bernoulli model studied here. However, our experiments in Section 3 show that an explicit thresholding layer also significantly improves the sample complexity of training a robust neural network on MNIST. We conjecture that the effectiveness of thresholding is behind many of the successful defenses against adversarial examples on MNIST (for instance, see Appendix C in [36]).

## 3    Experiments

We complement our theoretical results by performing experiments on multiple common datasets. We consider standard convolutional neural networks and train models on datasets of varying complexity. Specifically, we study the MNIST [34], CIFAR-10 [33], and SVHN [40] datasets. We use a simple convolutional architecture for MNIST, a standard ResNet model [23] for CIFAR-10, and a wider ResNet [61] for SVHN. We perform robust optimization to train our classifiers on perturbations generated by projected gradient descent. Appendix G provides additional details for our experiments.

**Empirical sample complexity evaluation.**    We study how the generalization performance of adversarially robust networks varies with the size of the training dataset. To do so, we train networks with a specific $\ell_\infty$ adversary while reducing the size of the training set. The training subsets are produced by randomly sub-sampling the complete dataset in a class-balanced fashion. When increasing the number of samples, we ensure that each dataset is a superset of the previous one.

We evaluate the robustness of each trained network to perturbations of varying magnitude ($\varepsilon_{test}$). For each choice of training set size $N$ and fixed attack $\varepsilon_{test}$, we select the best performance achieved across all hyperparameters settings (training perturbations $\varepsilon_{train}$ and model size). On all three datasets, we observed that the best standard accuracy is usually achieved for the standard trained network, while the best adversarial accuracy for almost all values of $\varepsilon_{test}$ was achieved when training with the largest $\varepsilon_{train}$. We maximize over the hyperparameter settings since we are not interested in the performance of a specific model, but rather in the inherent generalization properties of the dataset independently of the classifier used. Figure 2 shows the results of these experiments.

The plots demonstrate the need for more data to achieve adversarially robust generalization. For any fixed test set accuracy, the number of samples needed is significantly higher for robust generalization. In the SVHN experiments (where we have sufficient training samples to observe plateauing behavior), the standard accuracy reaches its maximum with significantly fewer samples than the adversarial accuracy. We report more details of our experiments in Section H of the supplementary material.

**Thresholding experiments.**    Motivated by our theoretical study of the Bernoulli model, we investigate whether thresholding can also improve the sample complexity of robust generalization against an $\ell_\infty$ adversary on MNIST.

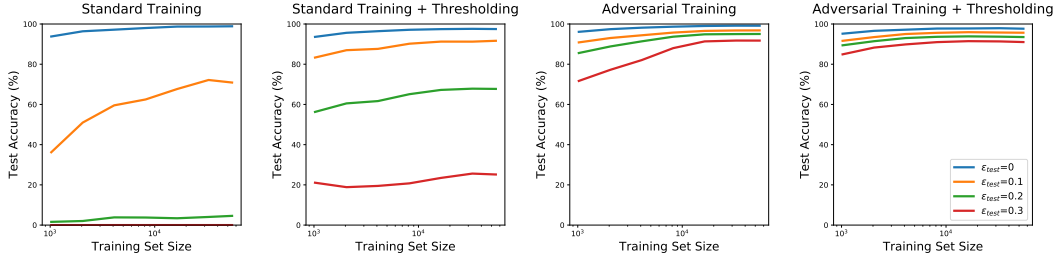

Figure 3: Adversarial robustness to $\ell_\infty$ attacks on the MNIST dataset for a simple convolution network [36] with and without explicit thresholding filters. For each training set size choice and $\varepsilon_{test}$, we report the best test set accuracy achieved over choice of thresholding filters and $\varepsilon_{train}$. We observe that introducing thresholding filters significantly reduces the number of samples needed to achieve good adversarial generalization.

We repeat the above sample complexity experiments with networks where thresholding filters are explicitly encoded in the model. Here, we replace the first convolutional layer with a fixed thresholding layer consisting of two channels, $\mathrm{ReLU}(x - \varepsilon_{filter})$ and $\mathrm{ReLU}(1 - x - \varepsilon_{filter})$, where $x$ is the input image. Figure 3 shows results for networks trained with such a thresholding layer. For standard trained networks, we use a value of $\varepsilon_{filter} = 0.1$ for the thresholding filters, whereas for adversarially trained networks we set $\varepsilon_{filter} = \varepsilon_{train}$. For each data subset size and test perturbation $\varepsilon_{test}$, we plot the best test accuracy achieved over networks trained with different thresholding filters, i.e., different values of $\varepsilon$. We separately show the effect of explicit thresholding in such networks when they are trained adversarially using PGD.

As predicted by our theory, the networks achieve good adversarially robust generalization with significantly fewer samples when thresholding filters are added. Further, note that adding a simple thresholding layer directly yields nearly state-of-the-art robustness against moderately strong adversaries ($\varepsilon = 0.1$), without any other modifications to the model architecture or training algorithm. It is also worth noting that the thresholding filters could have been learned by the original network architecture, and that this modification only decreases the capacity of the model. Our findings emphasize network architecture as a crucial factor for learning adversarially robust networks from a limited number of samples.

We also experimented with thresholding filters on the CIFAR10 dataset, but did not observe any significant difference from the standard architecture. This agrees with our theoretical understanding that thresholding helps primarily in the case of (approximately) binary datasets.

## 4 Related Work

Due to the large body of work on adversarial robustness, we focus on related papers that also provide theoretical explanations for adversarial examples. We defer a detailed discussion of related work to Appedix A and discuss here the works most closely related to ours.

Wang et al. [55] study the adversarial robustness of nearest neighbor classifiers. In contrast to our work, the authors give theoretical guarantees for a specific classification algorithm, and do not see a separation in sample complexity between robust and regular generalization. Recent work by Gilmer et al. [20] explores a specific distribution where robust learning is empirically difficult with overparametrized neural networks. The main phenomenon is that even a small natural error rate on their dataset translates to a large adversarial error rate. Our results give a more nuanced picture that involves the sample complexity required for generalization. In our data models, it is possible to achieve an error rate that is essentially zero by using a very small number of samples, whereas the adversarial error rate is still large unless we have seen a lot of samples.

The work of Xu et al. [58] establishes a connection between robust optimization and regularization for linear classification. In particular, they show that robustness to a specific perturbation set is exactly equivalent to the standard support vector machine. Subsequent work by Xu and Mannor [57] builds a deeper connection between robustness and generalization. They prove that for a certain notion of robustness, robust algorithms generalize. Moreover, they show that robustness is a necessary

condition for generalization in an asymptotic sense. Bellet and Habrard [5] gives similar results for metric learning. However, these results do no imply sample complexity bounds since they are asymptotic. Our results stand in stark contrast: we show that generalization can, in simple models, be significantly easier than robustness when sample complexity enters the picture.

Fawzi et al. [18] relate the robustness of linear and non-linear classifiers to adversarial and (semi-) random perturbations. Their work studies the setting where the classifier is fixed and does not encompass the learning task. Fawzi et al. [19] give provable lower bounds for adversarial robustness in models where robust classifiers do not exist. In contrast, we are interested in a setting where robust classifiers exist, but need many samples to learn. Papernot et al. [43] discuss adversarial robustness at the population level. We defer a more detailed discussion of these works to Appendix A.

There is also a long line of work in machine learning on exploring the connection between various notions of margin and generalization, e.g., see [46] and references therein. In this setting, the $\ell_p$ margin, i.e., how robustly classifiable the data is for $\ell_p^*$-bounded classifiers, enables dimension-independent control of the sample complexity. However, the sample complexity in concrete distributional models can often be significantly smaller than what the margin implies.

## 5 Discussion and Conclusions

The vulnerability of neural networks to adversarial perturbations has recently been a source of much discussion and is still poorly understood. Different works have argued that this vulnerability stems from their discontinuous nature [51], their linear nature [21], or is a result of high-dimensional geometry and independent of the model class [20]. Our work gives a more nuanced picture. We show that for a natural data distribution (the Gaussian model), the model class we train does not matter and a standard linear classifier achieves optimal robustness. However, robustness also strongly depends on properties of the underlying data distribution. For other data models (such as MNIST or the Bernoulli model), our results demonstrate that non-linearities are indispensable to learn from few samples. This dichotomy provides evidence that defenses against adversarial examples need to be tailored to the specific dataset (even for the same type of perturbations) and hence may be more complicated than a single, broad approach. Understanding the interactions between robustness, classifier model, and data distribution from the perspective of generalization is an important direction for future work. We refer the reader to Section B in the appendix for concrete questions in this direction.

The focus of our paper is on adversarial perturbations in a setting where the test distribution (before the adversary's action) is the same as the training distribution. While this is a natural scenario from a security point of view, other setups can be more relevant in different robustness contexts. For instance, we may want a classifier that is robust to small changes between the training and test distribution. This can be formalized as the classification accuracy on *unperturbed* examples coming from an *adversarially* modified distribution. Here, the power of the adversary is limited by how much the test distribution can be modified, and the adversary is not allowed to perturb individual samples coming from the modified test distribution. Interestingly, our lower bound for the Gaussian model also applies to such worst-case distributional shifts. In particular, if the adversary is allowed to shift the mean $\theta^\star$ by a vector in $\mathcal{B}_\infty^\varepsilon$, our proof sketched in Section C transfers to the distribution shift setting. Since the lower bound relies only on a single universal perturbation, this perturbation can also be applied directly to the mean vector.

What do our results mean for robust classification of real images? Our Gaussian lower bound implies that if an algorithm works for all (or most) settings of the unknown parameter $\theta^\star$, then achieving strong $\ell_\infty$-robustness requires a sample complexity increase that is polynomial in the dimension. There are a few different ways this lower bound could be bypassed. It is conceivable that the noise scale $\sigma$ is significantly smaller for real image datasets, making robust classification easier. Even if that was not the case, a good algorithm could work for the parameters $\theta^\star$ that correspond to real datasets while not working for most other parameters. To accomplish this, the algorithm would implicitly or explicitly have prior information about the correct $\theta^\star$. While some prior information is already incorporated in the model architectures (e.g., convolutional and pooling layers), the conventional wisdom usually is not to bias the neural network with our priors. Our work suggests that there are trade-offs with robustness here and that adding more prior information could help to learn more robust classifiers.

## Acknowledgements

During this research project, Ludwig Schmidt was supported by a Google PhD fellowship and a Microsoft Research fellowship at the Simons Institute for the Theory of Computing. Ludwig was also an intern in the Google Brain team. Shibani Santurkar is supported by the National Science Foundation (NSF) under grants IIS-1447786, IIS-1607189, and CCF-1563880, and the Intel Corporation. Dimitris Tsipras was supported in part by the NSF grant CCF-1553428 and the NSF Frontier grant CNS-1413920. Aleksander Mądry was supported in part by an Alfred P. Sloan Research Fellowship, a Google Research Award, and the NSF grants CCF-1553428 and CNS-1815221.

## Footnotes

[1]We remark that this accuracy is still currently the best published robust accuracy on CIFAR10 [3]. For instance, contemporary approaches to architecture tuning do not yield better robust accuracies [15].

[2]We write $\mathscr{P}(\mathbb{R}^d)$ to denote the power set of $\mathbb{R}^d$, i.e., the set of subsets of $\mathbb{R}^d$.

[3]We remark that it is also possible to study a more general setting where standard generalization requires a larger number of samples.

[4]To be precise, the two distributions have comparable sample complexity for standard generalization in the regime where $\sigma \approx \tau^{-1}$.

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
