[Supplementary Material]

# A   Related Work

Due to the large body of work on adversarial robustness, we focus on related papers that also provide theoretical explanations for adversarial examples. Compared to prior work, the main difference of our approach is the focus on generalization. Most related papers study robustness either without the learning context, or in the limit as the number of samples approaches infinity. As a result, finite sample phenomena do not arise in these theoretical approaches. As we have seen in Figure 1, adversarial examples are currently a failure of generalization from a limited training set. Hence we believe that studying robust generalization is an insightful avenue for understanding adversarial examples.

- Wang et al. [55] study the adversarial robustness of nearest neighbor classifiers. In contrast to our work, the authors give theoretical guarantees for a specific classification algorithm. We focus on the inherent sample complexity of adversarially robust generalization independently of the learning method. Moreover, our results hold for finite sample sizes while [55] compare the nearest neighbor classifier to the optimal classifier in the limit as $n \to \infty$.

- Recent work by Gilmer et al. [20] explores a specific distribution where robust learning is empirically difficult with overparametrized neural networks.[5] The main phenomenon is that even a small natural error rate on their dataset translates to a large adversarial error rate. Our results give a more nuanced picture that involves the sample complexity required for generalization. In our data models, it is possible to achieve an error rate that is essentially zero by using a very small number of samples, whereas the adversarial error rate is still large unless we have seen a lot of samples.

- Fawzi et al. [18] relate the robustness of linear and non-linear classifiers to adversarial and (semi-) random perturbations. Their work studies the setting where the classifier is fixed and does not encompass the learning task. We focus on generalization aspects of adversarial robustness and provide upper and lower bounds on the sample complexity. Overall, we argue that adversarial examples are inherent to the statistical setup and not necessarily a consequence of a concrete classifier model.

- The work of Xu et al. [58] establishes a connection between robust optimization and regularization for linear classification. In particular, they show that robustness to a specific perturbation set is exactly equivalent to the standard support vector machine. The authors give asymptotic consistency results under a robustness condition, but do not provide any finite sample guarantees. In contrast, our work considers specific distributional models where we can demonstrate a clear gap between robust and standard generalization.

- Papernot et al. [43] discuss adversarial robustness at the population level. They assume the existence of an adversary that can significantly increase the loss for *any* hypothesis in the hypothesis class. By definition, robustness against adversarial perturbations is impossible in this regime. As demonstrated in Figure 1, we instead conjecture that current classification models are not robust to adversarial examples because they fail to generalize. Hence our results concern generalization from a finite number of samples. We show that even when the hypothesis class is large enough to achieve good robust classification error, the sample complexity of robust generalization can still be significantly bigger than that of standard generalization.

- In a recent paper, Fawzi et al. [19] also give provable lower bounds for adversarial robustness. There are several important differences between their work and ours. At a high level, the results in [19] state that there are fundamental limits for adversarial robustness that apply to *any* classifier. As pointed out by the authors, their bounds also apply to the human visual system. However, an important aspect of adversarial examples is that they often fool current classifiers, yet are still easy to recognize for humans. Hence we believe that the approach in [19] does not capture the underlying phenomenon since it does not distinguish between the robustness of current artificial classifiers and the human visual system.

  Moreover, the lower bounds in [19] do not involve the training data and consequently apply in the limit where an infinite number of samples is available. In contrast, our work investigates how the amount of available training data affects adversarial robustness. As we have seen in Figure 1, adversarial robustness is currently an issue of *generalization*. In particular, we can train classifiers

that achieve a high level of robustness on the CIFAR10 training set, but this robustness does not transfer to the test set. Therefore, our perspective based on adversarially robust generalization more accurately reflects the current challenges in training robust classifiers.

Finally, Fawzi et al. [19] utilize the notion of a latent space for the data distribution in order to establish lower bounds that apply to any classifier. While the existence of generative models such as GANs provides empirical evidence for this assumption, we note that it does not suffice to accurately describe the robustness phenomenon. For instance, there are multiple generative models that produce high-quality samples for the MNIST dataset, yet there are now also several successful defenses against adversarial examples on MNIST. As we have shown in our work, the fine-grained properties of the data distribution can have significant impact on how hard it is to learn a robust classifier.

**Margin-based theory.** There is a long line of work in machine learning on exploring the connection between various notions of margin and generalization, e.g., see [46] and references therein. In this setting, the $\ell_p$ margin, i.e., how robustly classifiable the data is for $\ell_p^*$-bounded classifiers, enables dimension-independent control of the sample complexity. However, the sample complexity in concrete distributional models can often be significantly smaller than what the margin implies. As we will see next, standard margin-based bounds do not suffice to demonstrate a gap between robust and benign generalization for the distributional models studied in our work.

First, we briefly remind the reader about standard margin-based results (see Theorem 15.4 in [46] for details). For a dataset that has bounded $\ell_2$ norm $\rho$ and $\ell_2$ margin $\gamma$, the classification error of the hard-margin SVM scales as

$$\sqrt{\frac{(\rho/\gamma)^2}{n}}$$

where $n$ is the number of samples. To illustrate this bound, consider the Gaussian model in the regime $\sigma = \Theta(d^{1/4})$ where a single sample suffices to learn a classifier with low error (see Theorem 4). The standard bound on the norm of an i.i.d. Gaussian vector shows that we have a data norm bound $\rho = \Theta(d^{3/4})$ with high probability. While the Gaussian model is not strictly separable in any regime, we can still consider the probability that a sample achieves at least a certain margin:

$$\mathbb{P}_{z \sim \mathcal{N}(0, \sigma^2 I)} \left[ \frac{\langle z, \theta^\star \rangle}{\|\theta^\star\|_2} \geq \rho \right] \geq 1 - \delta \ .$$

A simple calculation shows that for $\|\theta^\star\|_2 = \sqrt{d}$ (as in our earlier bounds), the Gaussian model does not achieve margin $\gamma \geq \sqrt{d}$ even at the quantile $\delta = 1/2$. Hence the margin-based bound would indicate a sample complexity of $\Omega(d^{1/4})$ already for *standard* generalization, which obscures the dichotomy between standard and robust sample complexity.

**Robust statistics.** An orthogonal line of work in robust statistics studies robustness of estimators to corruption of *training* data [27]. This notion of robustness, while also important, is not directly relevant to the questions addressed in our work.

# B   Future Directions

Several questions remain. We now provide a list of concrete directions for future work on robust generalization.

**Stronger lower bounds.** An interesting aspect of adversarial examples is that the adversary can often fool the classifier on most inputs [11, 51]. While our results show a lower bound for classification error $1/2$, it is conceivable that misclassification rates much closer to 1 are unavoidable for at least one of the two classes (or equivalently, when the adversary is allowed to pick the class label). In order to avoid degenerate cases such as achieving robustness by being the constant classifier, it would be interesting to study regimes where the classifier has high standard accuracy but does not achieve robustness yet. In such a regime, does good standard accuracy imply that the classifier is vulnerable to adversarial perturbations on almost all inputs?

**Different perturbation sets.** Depending on the problem setting, different perturbation sets are relevant. Due to the large amount of empirical work on $\ell_\infty$ robustness, our paper has focused on such perturbations. From a security point of view, we want to defend against perturbations that are imperceptible to humans. While this is not a well-defined concept, the class of small $\ell_\infty$-norm perturbations should be contained in any reasonable definition of imperceptible perturbations. However, changes in different $\ell_p$ norms [11, 37, 51], sparse perturbations [10, 39, 42, 50], or mild spatial transformations can also be imperceptible to a human [56]. In less adversarial settings, more constrained and lower-dimensional perturbations such as small rotations and translations may be more appropriate [17]. Overall, understanding the sample complexity implications of different perturbation sets is an important direction for future work.

**Further notions of test time robustness.** As mentioned above, less adversarial forms of robustness may be better suited to model challenges arising outside security. How much easier is it to learn a robust classifier in more benign settings? This question is naturally related to problems such as transfer learning and domain adaptation.

**Broader classes of distributions.** Our results directly apply to two concrete distributional models. While the results already show interesting phenomena and are predictive of behavior on real data, understanding the robustness properties for a broader class of distributions is an important direction for future work. Moreover, it would be useful to understand what general properties of distributions make robust generalization hard or easy.

**Wider sample complexity separations.** In our work, we show a separation of $\sqrt{d}$ between the standard and robust sample complexity for the Gaussian model. It is open whether larger gaps are possible. Note that for large adversarial perturbations, the data may no longer be robustly separable which leads to trivial gaps in sample complexity, simply because the harder robust generalization problem is impossible to solve. Hence this question is mainly interesting in the regime where a robust classifier exists in the model class of interest.

**Robustness in the PAC model.** Our focus has been on robust learning for specific distributions without any limitations on the hypothesis class. A natural dual perspective is to investigate robust learning for specific hypothesis classes, as in the probably approximately correct (PAC) framework. For instance, it is well known that the sample complexity of learning a half space in $d$ dimensions is $O(d)$. Does this sample complexity also suffice to learn in the presence of an adversary at test time? While robustness to adversarial training noise has been studied in the PAC setting (e.g., see [9, 29, 30]), we are not aware of similar work on test time robustness.

## C  Lower Bounds for the Gaussian Model

Recall our main theoretical result: In the Gaussian model, no algorithm can produce a robust classifier unless it has seen a large number of samples. In particular, we give a nearly tight trade-off between the number of samples and the $\ell_\infty$-robustness of the classifier. We give a high level overview of the proof in this section, and provide additional technical details in the next section. The following theorem is the technical core of this lower bound. Combined with standard bounds on the $\ell_\infty$-norm of a random Gaussian vector, it gives Theorem 6.

**Theorem 11.** *Let $g_n$ be any learning algorithm, i.e., a function from $n$ samples in $\mathbb{R}^d \times \{\pm 1\}$ to a binary classifier $f_n$. Moreover, let $\sigma > 0$, let $\varepsilon \geq 0$, and let $\theta \in \mathbb{R}^d$ be drawn from $\mathcal{N}(0, I)$. We also draw $n$ samples from the $(\theta, \sigma)$-Gaussian model. Then the expected $\ell_\infty^\varepsilon$-robust classification error of $f_n$ is at least*

$$\frac{1}{2} \mathop{\mathbb{P}}_{v \sim \mathcal{N}(0, I)} \left[ \sqrt{\frac{n}{\sigma^2 + n}} \|v\|_\infty \leq \varepsilon \right].$$

Several remarks are in order. Since we lower bound the expected robust classification error for a distribution over the model parameters $\theta^\star$, our result implies a lower bound on the minimax robust classification error (i.e., minimum over learning algorithms, maximum over unknown parameters $\theta^\star$). Second, while we refer to the learning procedure as an algorithm, our lower bounds are information theoretic and hold irrespective of the computational power of this procedure.

Moreover, our proof shows that given the $n$ samples, there is a *single* adversarial perturbation that (a) applies to all learning algorithms, and (b) leads to at least a constant fraction of fresh samples being misclassified. In other words, the same perturbation is transferable across examples as well as across architectures and learning procedures. Hence our simple Gaussian data model already exhibits the transferability phenomenon, which has recently received significant attention in the deep learning literature (e.g., [38, 51, 52]).

We fix an algorithm $g_n$ and let $S_n$ denote the set of $n$ samples given to the algorithm. We are interested in the expected robust classification error, which can be formalized as

$$\mathbb{E}_{\theta^*}\mathbb{E}_{S_n}\mathbb{E}_{y\sim\pm 1}\Pr_{x\sim\mathcal{N}(y\theta^*,\sigma^2 I)}[\exists x'\in\mathcal{B}^\varepsilon_\infty(x):f_n(x')\neq y]\,.$$

We swap the two outer expectations so the quantity of interest becomes

$$\mathbb{E}_{S_n}\mathbb{E}_{\theta^*}\mathbb{E}_{y\sim\pm 1}\Pr_{x\sim\mathcal{N}(y\theta^*,\sigma^2 I)}[\exists x'\in\mathcal{B}^\varepsilon_\infty(x):f_n(x')\neq y]\,.$$

Given the samples $S_n$, the posterior on $\theta^*$ is a Gaussian distribution with parameters defined by simple statistics of $S_n$ (the sample mean and the number of samples). Since the new data point $x$ (to be classified) is itself drawn from a Gaussian distribution with mean $\theta^*$, the posterior distribution $\mu_+$ on the positive examples $x\sim\mathcal{N}(\theta^*,\sigma^2)$ is another Gaussian with a certain mean $\bar{z}$ and standard deviation $\sigma'$. Similarly, the posterior distribution $\mu_-$ on the negative examples is a Gaussian with mean $-\bar{z}$ and the same standard deviation $\sigma'$. At a high level, we will now argue that the adversary can make the two posterior distributions $\mu_-$ and $\mu_+$ similar enough so that the problem becomes inherently noisy, preventing any classifier from achieving a high accuracy.

To this end, define the classification sets of $f_n$ as $A_+=\{x\,|\,f_n(x)=+1\}$ and $A_-=\mathbb{R}^d\setminus A_+$. This allows us to write the expected robust classification error as

$$\mathbb{E}_{S_n}\mathbb{E}_{\theta^*}\left(\frac{1}{2}\Pr_{\mu_+}[\mathcal{B}^\varepsilon_\infty(A_-)]+\frac{1}{2}\Pr_{\mu_-}[\mathcal{B}^\varepsilon_\infty(A_+)]\right)\,.$$

We now lower bound the inner probabilities by considering the fixed perturbation $\Delta=\bar{z}$. Note that a point $x\sim\mu_+$ is certainly misclassified if we have $\|\Delta\|_\infty\leq\varepsilon$ and $x-\Delta\in A_-$. Thus the expected misclassification rate is at least $\mu_+(\{x\,|\,x-\Delta\in A_-\})=\mu_+(A_-+\Delta)$.[6] But since $\mu_+$ is simply a translated version of $N(0,\sigma'^2)$, this implies that

$$\Pr_{\mu_+}[\mathcal{B}^\varepsilon_\infty(A_-)]\ \geq\ \mu_0(A_-+\Delta-\bar{z})\ =\ \mu_0(A_-)$$

where the distribution $\mu_0$ is the centered Gaussian $\mu_0=\mathcal{N}(0,\sigma'^2)$. Similarly,

$$\Pr_{\mu_-}[\mathcal{B}^\varepsilon_\infty(A_+)]\ \geq\ \mu_0(A_+-\Delta+\bar{z})\ =\ \mu_0(A_+).$$

Since $\mu_0(A_-)+\mu_0(A_+)=1$, this implies that the adversarial perturbation $-\bar{z}$ misclassifies in expectation half of the positively labeled examples, which completes the proof. As mentioned above, the crucial step is that the posteriors $\mu_+$ and $\mu_-$ are similar enough so that we can shift both to the origin while still controlling the measure of the sets $A_-$ and $A_+$.

## D  Detailed proofs for the Gaussian model

### D.1  Upper bounds

We begin with standard results about (sub)-Gaussian concentration in Fact 12 and Lemmas 13 to 16. These results show that a class-weighted average of sufficiently many samples from the Gaussian model achieves a large inner product with the unknown mean vector. Lemma 17 then relates the inner product between a linear classifier and the mean vector to the classification accuracy. Theorem 18 uses the lemmas to establish our main theorem for standard generalization. Corollary 19 instantiates the bound for learning from one sample. After further simplification, this yields Theorem 4 from the main text.

For robust generalization, we first relate the inner product between a linear classifier and the unknown mean vector to the robust classification accuracy in Lemma 20. Similar to the standard classification error, Theorem 21 and Corollary 22 then yield our upper bounds for robust generalization. Simplifying Corollary 22 further gives Theorem 5 from the main text.

**Fact 12.** *Let $z \in \mathbb{R}^d$ be drawn from a centered spherical Gaussian, i.e., $z \sim \mathcal{N}_d(0, \sigma^2 I)$ where $\sigma > 0$. Then we have $\mathbb{P}[\|z\|_2 \geq \sigma\sqrt{d} + t] \leq e^{-t^2/(2\sigma^2)}$ .*

*Proof.* We refer the reader to Example 5.7 in [8] for a reference of this standard result. Combined with $\mathbb{E}[\|z\|_2] \leq \sigma\sqrt{d}$, which is obtained from Jensen's Inequality, the aforementioned example gives the desired upper tail bound. $\square$

**Lemma 13.** *Let $z_1, \ldots, z_n \in \mathbb{R}^d$ be drawn i.i.d. from a spherical Gaussian, i.e., $z_i \sim \mathcal{N}_d(\mu, \sigma^2 I)$ where $\mu \in \mathbb{R}^d$ and $\sigma > 0$. Let $\overline{z} \in \mathbb{R}^d$ be the sample mean vector $\overline{z} = \frac{1}{n}\sum_{i=1}^n z_i$. Finally, let $\delta > 0$ be the target probability. Then we have*

$$\mathbb{P}\left[ \|\overline{z}\|_2 \geq \|\mu\|_2 + \frac{\sigma\left(\sqrt{d} + \sqrt{2\log 1/\delta}\right)}{\sqrt{n}} \right] \leq \delta .$$

*Proof.* Since each $z_i$ has the same distribution as $\mu + g_i$ for $g_i \sim \mathcal{N}_d(0, \sigma^2 I)$, we can bound the desired tail probability for

$$\overline{z} = \frac{1}{n}\sum_{i=1}^n \mu + g_i$$
$$= \mu + \frac{1}{n}\sum_{i=1}^n g_i .$$

Morever, the average of the $g_i$ has the same distribution as $\overline{g} \sim \mathcal{N}_d(0, \frac{\sigma^2}{n}I)$. Hence it suffices to bound the tail of $\|\mu + \overline{g}\|_2$. For any $c \geq 0$, applying the triangle inequality then gives

$$\mathbb{P}[\|\overline{z}\|_2 \geq \|\mu\|_2 + c] = \mathbb{P}[\|\mu + \overline{g}\|_2 \geq \|\mu\|_2 + c]$$
$$\leq \mathbb{P}[\|\overline{g}\|_2 \geq c] .$$

Setting $c = \sigma\sqrt{d/n} + t$ with

$$t = \sigma\sqrt{\frac{2\log 1/\delta}{n}}$$

and substituting into Fact 12 then gives the desired result. $\square$

For convenient use in our later theorems, we instantiate Lemma 13 with the parameters most relevant for our Gaussian model. In particular, the norm of the mean vector $\mu$ is $\sqrt{d}$ and we are interested in up to exponentially small failure probability $\delta$ (but not necessarily smaller).

**Lemma 14.** *Let $z_1, \ldots, z_n \in \mathbb{R}^d$ be drawn i.i.d. from a spherical Gaussian with mean norm $\sqrt{d}$, i.e., $z_i \sim \mathcal{N}_d(\mu, \sigma^2 I)$ where $\mu \in \mathbb{R}^d$, $\|\mu\|_2 = \sqrt{d}$, and $\sigma > 0$. Let $\overline{z} \in \mathbb{R}^d$ be the sample mean vector $\overline{z} = \frac{1}{n}\sum_{i=1}^n z_i$. Then we have*

$$\mathbb{P}\left[ \|\overline{z}\|_2 \geq \left(1 + \frac{2\sigma}{\sqrt{n}}\right)\sqrt{d} \right] \leq e^{-d/2} .$$

*Proof.* We substitute into Lemma 13 with $\|\mu\|_2 = \sqrt{d}$ and $\sqrt{2\log 1/\delta} = \sqrt{d}$. $\square$

**Lemma 15.** *Let $z_1, \ldots, z_n \in \mathbb{R}^d$ be drawn i.i.d. from a spherical Gaussian, i.e., $z_i \sim \mathcal{N}_d(\mu, \sigma^2 I)$ where $\mu \in \mathbb{R}^d$ and $\sigma > 0$. Let $\overline{z} \in \mathbb{R}^d$ be the mean vector $\overline{z} = \frac{1}{n}\sum_{i=1}^n z_i$. Finally, let $\delta > 0$ be the target probability. Then we have*

$$\mathbb{P}\left[ \langle \overline{z}, \mu \rangle \leq \|\mu\|_2^2 - \sigma\|\mu\|_2\sqrt{\frac{2\log 1/\delta}{n}} \right] \leq \delta .$$

*Proof.* As in Lemma 13, we use the fact that $\bar{z}$ has the same distribution as $\mu + \bar{g}$ where $\bar{g} \sim \mathcal{N}_d(0, \frac{\sigma^2}{n}I)$. For any $t \geq 0$, this allows us to simplify the tail event to

$$\mathbb{P}\left[\langle \bar{z}, \mu \rangle \leq \|\mu\|_2^2 - t\right] = \mathbb{P}[\langle \bar{g}, \mu \rangle \leq -t] .$$

The right hand side can now be simplified to $\mathbb{P}[h \geq t]$ where $h \sim \mathcal{N}(0, \sigma^2\|\mu\|_2^2/n)$. Invoking the standard sub-Gaussian tail bound

$$\mathbb{P}[h \geq t] \leq \exp\left(-\frac{n \cdot t^2}{2\sigma^2\|\mu\|_2^2}\right)$$

and substituting $t = \sigma\|\mu\|_2\sqrt{\frac{2\log 1/\delta}{n}}$ then gives the desired result. $\qquad\square$

**Lemma 16.** *Let $z_1, \dots, z_n \in \mathbb{R}^d$ be drawn i.i.d. from a spherical Gaussian with mean norm $\sqrt{d}$, i.e., $z_i \sim \mathcal{N}_d(\mu, \sigma^2 I)$ where $\mu \in \mathbb{R}^d$, $\|\mu\|_2 = \sqrt{d}$, and $\sigma > 0$. Let $\bar{z} \in \mathbb{R}^d$ be the sample mean vector $\bar{z} = \frac{1}{n}\sum_{i=1}^n z_i$ and let $\widehat{w} \in \mathbb{R}^d$ be the unit vector in the direction of $\bar{z}$, i.e., $\widehat{w} = \bar{z}/\|\bar{z}\|_2$. Then we have*

$$\mathbb{P}\left[\langle \widehat{w}, \mu \rangle \leq \frac{2\sqrt{n}-1}{2\sqrt{n}+4\sigma}\sqrt{d}\right] \leq 2\exp\left(-\frac{d}{8(\sigma^2+1)}\right) .$$

*Proof.* We invoke Lemma 14, which yields

$$\|\bar{z}\|_2 \leq \left(1 + \frac{2\sigma}{\sqrt{n}}\right)\sqrt{d}$$

with probability $1 - e^{-d/2}$. Moreover, we invoke Lemma 15 with $\delta = e^{-d/8\sigma^2}$ and $\|\mu\|_2 = \sqrt{d}$ to get

$$\langle \bar{z}, \mu \rangle \geq d - \frac{d}{2\sqrt{n}}$$

with probability $1 - e^{-d/8\sigma^2}$. We continue under both events, which yields the desired overall failure probability $2e^{-d/2}$.

We now have

$$
\begin{aligned}
\langle \widehat{w}, \mu \rangle &= \frac{\langle \bar{z}, \mu \rangle}{\|\bar{z}\|_2} \\
&\geq \frac{\left(1 - \frac{1}{2\sqrt{n}}\right)d}{\|\bar{z}\|_2} \\
&\geq \frac{\left(1 - \frac{1}{2\sqrt{n}}\right)d}{\left(1 + \frac{2\sigma}{\sqrt{n}}\right)\sqrt{d}} \\
&= \frac{2\sqrt{n}-1}{2\sqrt{n}+4\sigma}\sqrt{d}
\end{aligned}
$$

as stated in the lemma. $\qquad\square$

**Lemma 17.** *Let $z \in \mathbb{R}^d$ be drawn from a spherical Gaussian, i.e., $z \sim \mathcal{N}_d(\mu, \sigma^2 I)$ where $\mu \in \mathbb{R}^d$ and $\sigma > 0$. Moreover, let $w \in \mathbb{R}^d$ be an arbitrary unit vector with $\langle w, \mu \rangle \geq \rho$ where $\rho \geq 0$. Then we have*

$$\mathbb{P}[\langle w, z \rangle \leq \rho] \leq \exp\left(-\frac{(\langle w, \mu \rangle - \rho)^2}{2\sigma^2}\right) .$$

*Proof.* Since $z$ has the same distribution as $\mu + g$ where $g \sim \mathcal{N}_d(0, \sigma^2 I)$, we can bound the tail event as

$$
\begin{aligned}
\mathbb{P}[\langle w, z \rangle \leq \rho] &= \mathbb{P}[\langle w, \mu + g \rangle \leq \rho] \\
&= \mathbb{P}[\langle w, g \rangle \leq \rho - \langle w, \mu \rangle] .
\end{aligned}
$$

The inner product $\langle w, g \rangle$ is distributed as a univariate normal $\mathcal{N}(0, \sigma^2)$ because the vector $w$ has unit norm. Hence we can invoke the standard sub-Gaussian tail bound to get the desired tail probability. $\qquad \square$

**Theorem 18** (Standard generalization in the Gaussian model.)**.** *Let $(x_1, y_1), \ldots, (x_n, y_n) \in \mathbb{R}^d \times \{\pm 1\}$ be drawn i.i.d. from a $(\theta^\star, \sigma)$-Gaussian model with $\|\theta^\star\|_2 = \sqrt{d}$. Let $\widehat{w} \in \mathbb{R}^d$ be the unit vector in the direction of $\overline{z} = \frac{1}{n} \sum_{i=1}^{n} y_i x_i$, i.e., $\widehat{w} = \overline{z}/\|\overline{z}\|_2$. Then with probability at least $1 - 2\exp(-\frac{d}{8(\sigma^2+1)})$, the linear classifier $f_{\widehat{w}}$ has classification error at most*

$$\exp\left(-\frac{(2\sqrt{n}-1)^2 d}{2(2\sqrt{n}+4\sigma)^2 \sigma^2}\right) \, .$$

*Proof.* Let $z_i = y_i \cdot x_i$ and note that each $z_i$ is independent and has distribution $\mathcal{N}_d(\theta^\star, \sigma^2 I)$. Hence we can invoke Lemma 16 and get

$$\langle \widehat{w}, \theta^\star \rangle \;\geq\; \frac{2\sqrt{n}-1}{2\sqrt{n}+4\sigma}\sqrt{d}$$

with probability at least $1 - 2\exp(-\frac{d}{8(\sigma^2+1)})$ as stated in the theorem.

Next, unwrapping the definition of $f_{\widehat{w}}$ allows us to write the classification error of $f_{\widehat{w}}$ as

$$\mathbb{P}[f_{\widehat{w}}(x) \neq y] \;=\; \mathbb{P}[\langle \widehat{w}, \theta^\star \rangle \leq 0] \, .$$

Invoking Lemma 17 with $\rho = 0$ then gives the desired bound. $\qquad \square$

**Corollary 19** (Generalization from a single sample.)**.** *Let $(x, y)$ be drawn from a $(\theta^\star, \sigma)$-Gaussian model with*

$$\sigma \;\leq\; \frac{d^{1/4}}{5\sqrt{\log 1/\beta}} \, .$$

*Let $\widehat{w} \in \mathbb{R}^d$ be the unit vector $\widehat{w} = \frac{yx}{\|x\|_2}$. Then with probability at least $1 - 2\exp(-\frac{d}{8(\sigma^2+1)})$, the linear classifier $f_{\widehat{w}}$ has classification error at most $\beta$.*

*Proof.* Invoking Theorem 18 with $n = 1$ gives a classification error bound of

$$\beta' \;=\; \exp\left(-\frac{d}{2(2+4\sigma)^2 \sigma^2}\right) \, .$$

It remains to show that $\beta' \leq \beta$.

We now bound the denominator in $\beta'$. First, we have

$$2 + 4\sigma \;\leq\; 2d^{1/4} + \frac{4}{5}d^{1/4}$$
$$\leq\; 3d^{1/4} \, .$$

Next, we bound the entire denominator as

$$2(2+4\sigma)^2 \sigma^2 \;\leq\; 2 \cdot 9\sqrt{d} \cdot \frac{\sqrt{d}}{25 \log 1/\beta}$$
$$\leq\; \frac{d}{\log 1/\beta}$$

which yields the desired classification error when substituted back into $\beta'$. $\qquad \square$

**Lemma 20.** *Assume a $(\theta^\star, \sigma)$-Gaussian model. Let $p \geq 1$, $\varepsilon \geq 0$ be robustness parameters, and let $\widehat{w}$ be a unit vector such that $\langle \widehat{w}, \theta^\star \rangle \geq \varepsilon \|\widehat{w}\|_p^*$, where $\|\cdot\|_p^*$ is the dual norm of $\|\cdot\|_p$. Then the linear classifier $f_{\widehat{w}}$ has $\ell_p^\varepsilon$-robust classification error at most*

$$\exp\left(-\frac{(\langle \widehat{w}, \theta^\star \rangle - \varepsilon\|\widehat{w}\|_p^*)^2}{2\sigma^2}\right) \, .$$

*Proof.* Per Definition 3, we have to upper bound the quantity

$$\mathop{\mathbb{P}}_{(x,y)\sim\mathcal{P}}\left[\exists\,x'\in\mathcal{B}(x)\,:\,f_{\widehat{w}}(x')\neq y\right].$$

For linear classifiers, we can rewrite this event as follows:

$$
\begin{aligned}
\mathop{\mathbb{P}}_{(x,y)\sim\mathcal{P}}\left[\exists\,x'\in\mathcal{B}_p^\varepsilon(x)\,:\,f_{\widehat{w}}(x')\neq y\right] &= \mathop{\mathbb{P}}_{(x,y)\sim\mathcal{P}}\left[\exists\,x'\in\mathcal{B}_p^\varepsilon(x)\,:\,\langle y\cdot x',\widehat{w}\rangle\leq 0\right]\\
&= \mathop{\mathbb{P}}_{(x,y)\sim\mathcal{P}}\left[\exists\,\Delta\in\mathcal{B}_p^\varepsilon(0)\,:\,\langle y\cdot(x+\Delta),\widehat{w}\rangle\leq 0\right]\\
&= \mathop{\mathbb{P}}_{(x,y)\sim\mathcal{P}}\left[\min_{\Delta\in\mathcal{B}_p^\varepsilon(0)}\langle y\cdot(x+\Delta),\widehat{w}\rangle\leq 0\right]\\
&= \mathop{\mathbb{P}}_{(x,y)\sim\mathcal{P}}\left[\langle y\cdot x,\widehat{w}\rangle+\min_{\Delta\in\mathcal{B}_p^\varepsilon(0)}\langle y\cdot\Delta,\widehat{w}\rangle\leq 0\right].
\end{aligned}
$$

We now use the definition of the dual norm. Note that for any $\Delta\in\mathcal{B}_p^\varepsilon$, we also have $-\Delta\in\mathcal{B}_p^\varepsilon$. Since $y\in\{\pm 1\}$, we can drop the $y$ factor. Overall, we get

$$
\begin{aligned}
\mathop{\mathbb{P}}_{(x,y)\sim\mathcal{P}}\left[\langle y\cdot x,\widehat{w}\rangle+\min_{\Delta\in\mathcal{B}_p^\varepsilon(0)}\langle y\cdot\Delta,\widehat{w}\rangle\leq 0\right] &= \mathop{\mathbb{P}}_{(x,y)\sim\mathcal{P}}\left[\langle y\cdot x,\widehat{w}\rangle-\varepsilon\|\widehat{w}\|_p^*\leq 0\right]\\
&= \mathop{\mathbb{P}}_{(x,y)\sim\mathcal{P}}\left[\langle y\cdot x,\widehat{w}\rangle\leq\varepsilon\|\widehat{w}\|_p^*\right].
\end{aligned}
$$

By assumption in the lemma, we have $\langle\widehat{w},\theta^\star\rangle\geq\varepsilon\|\widehat{w}\|_p^*$. Hence we can invoke Lemma 17 with $\mu=\theta^\star$ and $\rho=\varepsilon\|\widehat{w}\|_p^*$ to get the desired bound on the robust classification error. $\square$

**Theorem 21.** *Let* $(x_1,y_1),\ldots,(x_n,y_n)\in\mathbb{R}^d\times\{\pm 1\}$ *be drawn i.i.d. from a* $(\theta^\star,\sigma)$-*Gaussian model with* $\|\theta^\star\|_2=\sqrt{d}$. *Let* $\widehat{w}\in\mathbb{R}^d$ *be the unit vector in the direction of* $\overline{z}=\frac{1}{n}\sum_{i=1}^n y_i x_i$, *i.e.,* $\widehat{w}=\overline{z}/\|\overline{z}\|_2$. *Then with probability at least* $1-2\exp(-\frac{d}{8(\sigma^2+1)})$, *the linear classifier* $f_{\widehat{w}}$ *has* $\ell_\infty^\varepsilon$-*robust classification error at most* $\beta$ *if*

$$\varepsilon\;\leq\;\frac{2\sqrt{n}-1}{2\sqrt{n}+4\sigma}-\frac{\sigma\sqrt{2\log 1/\beta}}{\sqrt{d}}\;.$$

*Proof.* Let $z_i=y_i\cdot x_i$ and note that each $z_i$ is independent and has distribution $\mathcal{N}_d(\theta^\star,\sigma^2 I)$. Hence we can invoke Lemma 16 and get

$$\langle\widehat{w},\theta^\star\rangle\;\geq\;\frac{2\sqrt{n}-1}{2\sqrt{n}+4\sigma}\sqrt{d}$$

with probability at least $1-2\exp(-\frac{d}{8(\sigma^2+1)})$ as stated in the theorem.

Since $\|\widehat{w}\|_2=1$, we have $\|\widehat{w}\|_\infty^*=\|\widehat{w}\|_1\leq\sqrt{d}$. The bound on $\varepsilon$ in the theorem allows us to invoke Lemma 20. This yields an $\ell_2^\varepsilon$-robust classification error of at most

$$\exp\left(-\frac{(\langle\widehat{w},\theta^\star\rangle-\varepsilon\sqrt{d})^2}{2\sigma^2}\right).$$

Since

$$\langle\widehat{w},\theta^\star\rangle-\varepsilon\|\widehat{w}\|_p^*\;\geq\;\frac{2\sqrt{n}-1}{2\sqrt{n}+4\sigma}\sqrt{d}-\sqrt{d}\left(\frac{2\sqrt{n}-1}{2\sqrt{n}+4\sigma}-\frac{\sigma\sqrt{2\log 1/\beta}}{\sqrt{d}}\right)$$

this simplifies to the robust classification error stated in the theorem. $\square$

**Corollary 22.** *Let* $(x_1,y_1),\ldots,(x_n,y_n)\in\mathbb{R}^d\times\{\pm 1\}$ *be drawn i.i.d. from a* $(\theta^\star,\sigma)$-*Gaussian model with* $\|\theta^\star\|_2=\sqrt{d}$ *and* $\sigma\leq\frac{1}{32}d^{1/4}$. *Let* $\widehat{w}\in\mathbb{R}^d$ *be the unit vector in the direction of* $\overline{z}=\frac{1}{n}\sum_{i=1}^n y_i x_i$, *i.e.,* $\widehat{w}=\overline{z}/\|\overline{z}\|_2$. *Then with probability at least* $1-2\exp(-\frac{d}{8(\sigma^2+1)})$, *the linear classifier* $f_{\widehat{w}}$ *has* $\ell_\infty^\varepsilon$-*robust classification error at most* $0.01$ *if*

$$n\;\geq\;\begin{cases}1 & \text{for }\;\varepsilon\leq\frac{1}{4}d^{-1/4}\\64\,\varepsilon^2\sqrt{d} & \text{for }\;\frac{1}{4}d^{-1/4}\leq\varepsilon\leq\frac{1}{4}\end{cases}.$$

*Proof.* We begin by invoking Theorem 21, which gives a $\ell_\infty^{\varepsilon'}$-robust classification error at most $\beta = 0.01$ for

$$\varepsilon' = \frac{2\sqrt{n}-1}{2\sqrt{n}+4\sigma} - \frac{\sigma\sqrt{2\log{1/\beta}}}{\sqrt{d}}$$

$$\geq \frac{2\sqrt{n}-1}{2\sqrt{n}+\frac{1}{8}d^{1/4}} - \frac{1}{8d^{1/4}} \; .$$

First, we consider the case where $\varepsilon \leq \frac{1}{4}d^{-1/4}$. Using $n=1$, the resulting robustness is

$$\varepsilon' \geq \frac{1}{2+\frac{1}{8}d^{1/4}} - \frac{1}{8d^{1/4}}$$

$$\geq \frac{1}{(2+\frac{1}{8})d^{1/4}} - \frac{1}{8d^{1/4}}$$

$$\geq \frac{1}{4}d^{-1/4}$$

$$\geq \varepsilon$$

as required.

Next, we consider the case $\frac{1}{4}d^{-1/4} \leq \varepsilon \leq \frac{1}{4}$. Substituting $n = 64\varepsilon^2\sqrt{d}$, we get

$$\varepsilon' \geq \frac{16\varepsilon d^{1/4}-1}{16\varepsilon d^{1/4}+\frac{1}{8}d^{1/4}} - \frac{1}{8d^{1/4}}$$

$$\geq \frac{12\varepsilon d^{1/4}}{4d^{1/4}+\frac{1}{8}d^{1/4}} - \frac{1}{8d^{1/4}}$$

$$\geq \frac{12}{5}\varepsilon - \frac{1}{2}\varepsilon$$

$$\geq \varepsilon$$

which completes the proof. $\qquad\square$

### D.2 Lower bound

The following theorem is our main lower bound for the Gaussian model. To make the lower bound easily comparable to Corollary 22 on the upper bound side, we simplify the lower bound in Corollary 23 and bring it into a similar form.

**Theorem 11.** *Let $g_n$ be any learning algorithm, i.e., a function from $n$ samples in $\mathbb{R}^d \times \{\pm 1\}$ to a binary classifier $f_n$. Moreover, let $\sigma > 0$, let $\varepsilon \geq 0$, and let $\theta \in \mathbb{R}^d$ be drawn from $\mathcal{N}(0,I)$. We also draw $n$ samples from the $(\theta,\sigma)$-Gaussian model. Then the expected $\ell_\infty^\varepsilon$-robust classification error of $f_n$ is at least*

$$\frac{1}{2}\mathop{\mathbb{P}}_{v\sim\mathcal{N}(0,I)}\left[\sqrt{\frac{n}{\sigma^2+n}}\|v\|_\infty \leq \varepsilon\right] \; .$$

*Proof.* We begin by formally stating the expected $\ell_\infty^\varepsilon$-robust classification error of $f_n$:

$$\Xi = \mathop{\mathbb{E}}_{\theta\sim\mathcal{N}(0,I)}\left[\mathop{\mathbb{E}}_{\substack{y_1,\dots,y_n\sim\mathcal{R}}}\left[\mathop{\mathbb{E}}_{\substack{x_1,\dots,x_n\\\sim\mathcal{N}(y_i\theta,\sigma^2 I)}}\left[\mathop{\mathbb{E}}_{y\sim\mathcal{R}}\left[\mathop{\mathbb{P}}_{x\sim\mathcal{N}(y\theta,\sigma^2 I)}[\exists\, x' \in \mathcal{B}_\infty^\varepsilon(x) \,:\, f_n(x') \neq y]\right]\right]\right]\right]$$

where it is important to note that $f_n = g_n((x_1,y_1),\dots,(x_n,y_n))$ depends on the samples $(x_i, y_i)$ but not on $\theta$. This will allow us to re-arrange the above expectations in a crucial way.

We first rewrite the expectations by noting that we can sample $z_i \sim \mathcal{N}(\theta, \sigma^2 I)$ without conditioning on the class $y_i$ by then setting $f_n = g_n((y_1 z_1, y_1), \ldots, (y_n z_n, y_n))$. This yields

$$
\begin{aligned}
\Xi \;&=\; \mathop{\mathbb{E}}_{\theta \sim \mathcal{N}(0,I)} \left[ \mathop{\mathbb{E}}_{y_1,\ldots,y_n \sim \mathcal{R}} \left[ \mathop{\mathbb{E}}_{\substack{z_1,\ldots,z_n \\ \sim \mathcal{N}(\theta,\sigma^2 I)}} \left[ \mathop{\mathbb{E}}_{y \sim \mathcal{R}} \left[ \mathop{\mathbb{P}}_{x \sim \mathcal{N}(y\theta,\sigma^2 I)} \left[ \exists\, x' \in \mathcal{B}_\infty^\varepsilon(x) \,:\, f_n(x') \neq y \right] \right] \right] \right] \right] \\[4pt]
&=\; \mathop{\mathbb{E}}_{y_1,\ldots,y_n \sim \mathcal{R}} \left[ \mathop{\mathbb{E}}_{\theta \sim \mathcal{N}(0,I)} \left[ \mathop{\mathbb{E}}_{\substack{z_1,\ldots,z_n \\ \sim \mathcal{N}(\theta,\sigma^2 I)}} \left[ \mathop{\mathbb{E}}_{y \sim \mathcal{R}} \left[ \mathop{\mathbb{P}}_{x \sim \mathcal{N}(y\theta,\sigma^2 I)} \left[ \exists\, x' \in \mathcal{B}_\infty^\varepsilon(x) \,:\, f_n(x') \neq y \right] \right] \right] \right] \right]
\end{aligned}
$$

where in the second line we moved the expectation over the class labels to the outside.

Next, we will swap the order of the expectations over the mean parameter $\theta$ and the conditional samples $x_i$. Since the posterior distribution for a Gaussian prior and likelihood is also Gaussian, the conditional distribution of $\theta$ given the $z_i$ is a multivariate Gaussian with parameters

$$
\begin{aligned}
\mu' \;&=\; \frac{n}{\sigma^2 + n}\,\overline{z} \\[4pt]
\Sigma' \;&=\; \frac{\sigma^2}{\sigma^2 + n}\, I
\end{aligned}
$$

where $\overline{z} = \sum_{i=1}^n z_i$. Moreover, let $\mathcal{M}$ be the marginal distribution over $(z_i, \ldots, z_n)$ after integrating over $\theta$ (which we will analyze later). Then we get

$$
\Xi \;=\; \mathop{\mathbb{E}}_{y_1,\ldots,y_n \sim \mathcal{R}} \left[ \mathop{\mathbb{E}}_{(z_1,\ldots,z_n) \sim \mathcal{M}} \left[ \underbrace{\mathop{\mathbb{E}}_{\theta \sim \mathcal{N}(\mu',\Sigma')} \left[ \mathop{\mathbb{E}}_{y \sim \mathcal{R}} \left[ \mathop{\mathbb{P}}_{x \sim \mathcal{N}(y\theta,\sigma^2 I)} \left[ \exists\, x' \in \mathcal{B}_\infty^\varepsilon(x) \,:\, f_n(x') \neq y \right] \right] \right]}_{=\Psi} \right] \right]
\tag{1}
$$

We now bound the term $\Psi$. Since the inner events only depends on $\theta$ through $x$, we can combine the Gaussian expectation with the Gaussian probability after moving the expectation over the label $y$ to the outside. This gives

$$
\begin{aligned}
\Psi \;&=\; \mathop{\mathbb{E}}_{\theta \sim \mathcal{N}(\mu',\Sigma')} \left[ \mathop{\mathbb{E}}_{y \sim \mathcal{R}} \left[ \mathop{\mathbb{P}}_{x \sim \mathcal{N}(y\theta,\sigma^2 I)} \left[ \exists\, x' \in \mathcal{B}_\infty^\varepsilon(x) \,:\, f_n(x') \neq y \right] \right] \right] \\[4pt]
&=\; \mathop{\mathbb{E}}_{y \sim \mathcal{R}} \left[ \mathop{\mathbb{E}}_{\theta \sim \mathcal{N}(\mu',\Sigma')} \left[ \mathop{\mathbb{P}}_{x \sim \mathcal{N}(y\theta,\sigma^2 I)} \left[ \exists\, x' \in \mathcal{B}_\infty^\varepsilon(x) \,:\, f_n(x') \neq y \right] \right] \right] \\[4pt]
&=\; \mathop{\mathbb{E}}_{y \sim \mathcal{R}} \left[ \mathop{\mathbb{P}}_{x \sim \mathcal{N}(y\mu',\Sigma'')} \left[ \exists\, x' \in \mathcal{B}_\infty^\varepsilon(x) \,:\, f_n(x') \neq y \right] \right]
\end{aligned}
\tag{2}
$$

where $\Sigma'' = \Sigma' + \sigma^2 I$.

Next, we bound the $y = +1$ case in the expectation over $y$. The $y = -1$ case can be handled exactly analogously. We introduce the set $A_- \subseteq \mathbb{R}^d$ as the set of inputs on which the classifier $f_n$ returns $-1$, i.e., $A_- = \{x \mid f_n(x) = -1\}$. Note that we can treat $A_-$ as fixed here since it only depends on the samples $z_i$ and labels $y_i$ but not on the parameter $\theta$ or the new sample $x$. This allows us to rewrite the first event as

$$
\begin{aligned}
\{x \mid \exists\, x' \in \mathcal{B}_\infty^\varepsilon(x) \,:\, f_n(x') \neq +1\} \;&=\; \{x \mid \exists\, x' \in A_- \,:\, \|x - x'\|_\infty \leq \varepsilon\} \\
&=\; \mathcal{B}_\infty^\varepsilon(A_-)\,.
\end{aligned}
$$

Now, note that as long as $\|\mu'\|_\infty \leq \varepsilon$, the set $\mathcal{B}_\infty^\varepsilon(A_-)$ contains a copy of $A_-$ shifted by $\pm \mu'$. Hence we have

$$
\begin{aligned}
\mathop{\mathbb{P}}_{x \sim \mathcal{N}(\mu',\Sigma'')} \left[ \exists\, x' \in \mathcal{B}_\infty^\varepsilon(x) \,:\, f_n(x') \neq +1 \right] \;&=\; \mathop{\mathbb{P}}_{\mathcal{N}(\mu',\Sigma'')} \left[ \mathcal{B}_\infty^\varepsilon(A_-) \right] \\[4pt]
&\geq\; \mathbb{I}[\|\mu'\|_\infty \leq \varepsilon] \cdot \mathop{\mathbb{P}}_{\mathcal{N}(0,\Sigma'')} \left[ A_- \right]
\end{aligned}
$$

Repeating the same argument for the $y = -1$ case and substituting back into Equation (2) yields

$$\Psi \geq \mathop{\mathbb{E}}_{y \sim \mathcal{R}} \left[ \mathbb{I}[\|\mu'\|_\infty \leq \varepsilon] \cdot \mathop{\mathbb{P}}_{\mathcal{N}(0,\Sigma'')} \left[ A_{-\operatorname{sgn}(y)} \right] \right]$$

$$= \mathbb{I}[\|\mu'\|_\infty \leq \varepsilon] \cdot \frac{1}{2} \left( \mathop{\mathbb{P}}_{\mathcal{N}(0,\Sigma'')} [A_-] + \mathop{\mathbb{P}}_{\mathcal{N}(0,\Sigma'')} [A_+] \right)$$

$$= \frac{1}{2} \mathbb{I}[\|\mu'\|_\infty \leq \varepsilon] .$$

In the last line, we used that the sets two sets $A_-$ and $A_+$ are complements of each other and hence their total mass under the measure $\mathcal{N}(0, \Sigma'')$ is 1.

Substituting back into Equation (1) yields

$$\Xi \geq \mathop{\mathbb{E}}_{y_1,\ldots,y_n \sim \mathcal{R}} \left[ \mathop{\mathbb{E}}_{(z_1,\ldots,z_n) \sim \mathcal{M}} \left[ \frac{1}{2} \mathbb{I}[\|\mu'\|_\infty \leq \varepsilon] \right] \right]$$

$$= \frac{1}{2} \mathop{\mathbb{E}}_{(z_1,\ldots,z_n) \sim \mathcal{M}} [\mathbb{I}[\|\mu'\|_\infty \leq \varepsilon]]$$

$$= \frac{1}{2} \mathop{\mathbb{P}}_{(z_1,\ldots,z_n) \sim \mathcal{M}} \left[ \frac{n}{\sigma^2 + n} \|\bar{z}\|_\infty \leq \varepsilon \right]$$

where we dropped the expectation over the labels $y_i$ since the inner expression is now independent of the labels.

It remains to analyze the distribution of the vector $\bar{z}$. Note that conditioned on a vector $\theta_2 \sim \mathcal{N}_d(0, I)$, the distribution of each $z_i$ is $\mathcal{N}(\theta_2, \sigma^2 I)$. Hence the conditional distribution of $\bar{z}$ given $\theta_2$ is $\mathcal{N}(\theta_2, \frac{\sigma^2}{n} I)$ and integrating over $\theta_2$ yields a marginal distribution of $\mathcal{N}(0, (1 + \frac{\sigma^2}{n}) I)$. Overall, this gives

$$\Xi \geq \frac{1}{2} \mathop{\mathbb{P}}_{\theta_2 \sim \mathcal{N}(0,(1+\frac{\sigma^2}{n})I)} \left[ \frac{n}{\sigma^2 + n} \|\theta_2\|_\infty \leq \varepsilon \right]$$

$$= \frac{1}{2} \mathop{\mathbb{P}}_{\theta_2 \sim \mathcal{N}(0,I)} \left[ \sqrt{\frac{n}{\sigma^2 + n}} \|\theta_2\|_\infty \leq \varepsilon \right]$$

where we used

$$\frac{n}{\sigma^2 + n} \sqrt{1 + \frac{\sigma^2}{n}} = \sqrt{\frac{n}{\sigma^2 + n}} .$$

Rearranging this inequality yields the statement of the theorem. $\qquad \square$

**Corollary 23.** *Let $g_n$ be any learning algorithm, i.e., a function from $n \geq 0$ samples in $\mathbb{R}^d \times \{\pm 1\}$ to a binary classifier $f_n$. Moreover, let $\sigma > 0$, let $\varepsilon \geq 0$, and let $\theta \in \mathbb{R}^d$ be drawn from $\mathcal{N}(0, I)$. We also draw $n$ samples from the $(\theta, \sigma)$-Gaussian model. Then the expected $\ell_\infty^\varepsilon$-robust classification error of $f_n$ is at least $(1 - 1/d)\frac{1}{2}$ if*

$$n \leq \frac{\varepsilon^2 \sigma^2}{8 \log d} .$$

*Proof.* We have

$$\sqrt{\frac{n}{\sigma^2 + n}} \leq \sqrt{\frac{\varepsilon^2 \sigma^2}{8\sigma^2 \log d}} = \frac{\varepsilon}{2\sqrt{2 \log d}} .$$

Hence we get

$$\mathop{\mathbb{P}}_{v \sim \mathcal{N}(0,I)} \left[ \sqrt{\frac{n}{\sigma^2 + n}} \|v\|_\infty \leq \varepsilon \right] \geq \mathop{\mathbb{P}}_{v \sim \mathcal{N}(0,I)} \left[ \sqrt{\frac{\varepsilon}{2\sqrt{2 \log d}}} \|v\|_\infty \leq \varepsilon \right]$$

$$= \mathop{\mathbb{P}}_{v \sim \mathcal{N}(0,I)} \left[ \|v\|_\infty \leq 2\sqrt{2 \log d} \right] .$$

Standard concentration results for the maximum of $d$ i.i.d. Gaussians (e.g., see Theorem 5.8 in [8]) now imply that the above probability is at least $(1 - 1/d)$. Invoking into Theorem 11 then completes the proof of this corollary. $\qquad \square$

# E  Lower Bounds for the Bernoulli Model

For the Bernoulli model, our lower bound applies only to *linear* classifiers. As pointed out in Section 2.2, non-linear classifiers do not suffer an increase in sample complexity in this data model. We now give a high-level overview of our proof that the sample complexity lower bound. The next section provides a complete proof.

$$n \geq c \, \frac{\varepsilon^2 d}{\log d} \, . \tag{3}$$

At first, this lower bound may look stronger than in the Gaussian case, where Theorem 6 established a lower bound of the form $\frac{\varepsilon^2 \sqrt{d}}{\log d}$, i.e., with only a square root dependence on $d$. However, it is important to note that the relevant $\ell_\infty$-robustness scale for linear classifiers in the Bernoulli model is on the order of $\Theta(\tau)$, whereas non-linear classifiers can achieve robustness for noise level $\varepsilon$ up to 1. In particular, we prove that no linear classifier can achieve small $\ell_\infty$-robust classification error for $\varepsilon > 3\tau$ (see Lemma 30 in Appendix F.2 for details). Recall that we focus on the $\tau = \Theta(d^{-\frac{1}{4}})$ regime. In this case, the lower bound in Equation 3 is on the order of $\sqrt{d}$ samples, which is comparable to the (nearly) tight bound for the Gaussian case. This is no coincidence: for our noise parameters $\sigma \approx \tau^{-1} \approx d^{\frac{1}{4}}$, one can show that approximately $\sigma^2 = \sqrt{d}$ samples suffice to recover $\theta^*$ to sufficiently good accuracy.

The point of start of our proof of the lower bound for linear classifiers is the following observation. For an example $(x, y)$, a linear classifier with parameter vector $w$ robustly classifies the point $x$ if and only if

$$\inf_{\Delta : \|\Delta\|_\infty \leq \varepsilon} \langle yw, x + \Delta \rangle \; > \; 0 \, ,$$

which is equivalent to

$$\langle yw, x \rangle \; > \; \sup_{\Delta : \|\Delta\|_\infty \leq \varepsilon} \langle yw, \Delta \rangle \, .$$

By the definition of dual norms, the supremum on the right hand size is thus equal to $\varepsilon \|yw\|_1 = \varepsilon \|w\|_1$.

The learning algorithm infers the parameter vector $w$ from a limited number of samples. Since these samples are noisy copies of the unknown parameters $\theta^\star$, the algorithm cannot be too certain of any single bit in $\theta^*$ (recall that we draw $\theta^*$ uniformly from the hypercube). We formalize this intuition in Lemma 29 (Appendix F.2) as a bound on the log odds given a sample $S$:

$$\log \frac{\Pr[\theta = +1 \mid S]}{\Pr[\theta = -1 \mid S]} \, .$$

Given such a bound, we can analyze the uncertainty in the estimate $w$ by establishing an upper bound on the posterior $|\mathbb{E}[\theta_i^\star | S]|$ for each $i \in [d]$. This in turn allow us to bound $\mathbb{E}[\langle w, \theta^\star \rangle | S]$. With control over this expectation, we can then relate the prediction $\langle w, x \rangle$ and the $\ell_1$-norm $\|w\|_1$ via a tail bound argument.

# F  Detailed proofs for the Bernoulli model

## F.1  Upper bounds

As in the Gaussian case, our upper bounds rely on standard sub-Gaussian concentration. Lemmas 24 and 25 provide lower bounds on the inner product between a single sample from the Bernoulli model and the unknown parameter vectors. Lemma 26 then relates the inner product between a linear classifier and the unknown mean vector to the classification accuracy. Combining these results yields Theorem 27 for generalization from a single sample. Simplifying this theorem yields Corollary 28, which directly implies Theorem 8 from the main text.

**Lemma 24.** *Let $(x, y) \in \mathbb{R}^d \times \{\pm 1\}$ be a sample drawn from a $(\theta^\star, \tau)$-Bernoulli model and let $z = xy$. Let $\delta > 0$ be the target probability. Then we have*

$$\mathbb{P}\left[ \langle z, \theta^\star \rangle \; \leq \; 2\tau d - \sqrt{2d \log 1/\delta} \right] \; \leq \; \delta \, .$$

*Proof.* To center $z$, we define $g = z - \mathbb{E}[z] = z - 2\tau\theta^\star$, where each coordinate of $g$ has zero mean. Then, we can write

$$\langle z, \theta^\star \rangle = \langle g + 2\tau\theta^\star, \theta^\star \rangle = \langle g, \theta^\star \rangle + 2\tau d \,.$$

Hence for all $t > 0$ we have,

$$\mathbb{P}[\langle z, \theta^\star \rangle \leq 2\tau d - t] = \mathbb{P}[\langle g, \theta^\star \rangle \leq -t] \,.$$

Note that $g = (g_1, g_2, ..., g_d)$ is a vector of sub-Gaussian random variables since each entry is bounded, i.e., each $g_j$ (like $z_j$) lies in an interval of length 2. Hence, the sub-Gaussian parameter of each $g_j$ is 1. Invoking Corollary 1.7 from Rigollet and Hütter [45] for the weighted combination of independent sub-Gaussian random variables, we get that

$$\mathbb{P}\left[\sum_{j=1}^{d} g_j \theta_j^\star \leq -t\right] \leq \exp\left(-\frac{t^2}{2||\theta^\star||_2^2}\right) \,.$$

Since $||\theta^\star||_2^2 = d$, we can simplify the tail event

$$\mathbb{P}[\langle z, \theta^\star \rangle \leq 2\tau d - t] \leq \exp\left(-\frac{t^2}{2d}\right)$$

which then gives

$$\mathbb{P}\left[\langle z, \theta^\star \rangle \leq 2\tau d - \sqrt{2d \log 1/\delta}\right] \leq \delta$$

as desired. $\qquad\square$

**Lemma 25.** *Let* $(x, y) \in \mathbb{R}^d \times \{\pm 1\}$ *be a sample drawn from a* $(\theta^\star, \tau)$*-Bernoulli model and let* $z = xy$. *Let* $\widehat{w} \in \mathbb{R}^d$ *be the unit vector in the direction of $z$, i.e.,* $\widehat{w} = z/||z||_2$. *Then we have*

$$\mathbb{P}\left[\langle \widehat{w}, \theta^\star \rangle \leq \tau\sqrt{d}\right] \leq \exp\left(-\frac{\tau^2 d}{2}\right) \,.$$

*Proof.* We know that

$$||z||_2 = \sqrt{d} \,.$$

Moreover, we invoke Lemma 24 with $\delta = \exp(-\frac{\tau^2 d}{2})$ to get

$$\langle z, \theta^\star \rangle \leq \tau d$$

with probability $\delta$. We now have

$$\langle \widehat{w}, \theta^\star \rangle = \frac{\langle z, \theta^\star \rangle}{||z||_2}$$
$$\leq \frac{\tau d}{\sqrt{d}}$$

with probability $\delta$ as stated in the lemma. $\qquad\square$

**Lemma 26.** *Let* $(x, y) \in \mathbb{R}^d \times \{\pm 1\}$ *be a sample drawn from a* $(\theta^\star, \tau)$*-Bernoulli model and let* $z = xy$. *Moreover, let* $w \in \mathbb{R}^d$ *be an arbitrary unit vector with* $\langle w, 2\tau\theta^\star \rangle \geq 0$. *Then we have*

$$\mathbb{P}[\langle w, z \rangle \leq 0] \leq \exp\left(-2\tau^2 \langle w, \theta^\star \rangle^2\right) \,.$$

*Proof.* As in Lemma 24, we center $z = 2\tau\theta^\star + g$, where $g$ is a vector of zero-mean sub-Gaussian random variables. We can bound the tail event as

$$\mathbb{P}[\langle w, z \rangle \leq 0] = \mathbb{P}[\langle w, 2\tau\theta^\star + g \rangle \leq 0]$$
$$= \mathbb{P}[\langle w, g \rangle \leq -\langle w, 2\tau\theta^\star \rangle] \,.$$

We know that the sub-Gaussian parameter of each $g_j$ is 1 as discussed in Lemma 24. Hence, invoking Corollary 1.7 from Rigollet and Hütter [45] for the weighted combination of independent sub-gaussian random variables, we get that

$$\mathbb{P}\left[\sum_{j=1}^{d} g_j w_j \leq -t\right] \leq \exp\left(-\frac{t^2}{2||w||_2^2}\right) = \exp\left(-\frac{t^2}{2}\right).$$

Thus,

$$\mathbb{P}[\langle w, g\rangle \leq -\langle w, 2\tau\theta^\star\rangle] \leq \exp\left(-\frac{\langle w, 2\tau\theta^\star\rangle^2}{2}\right)$$

as desired in the lemma. $\qquad\square$

**Theorem 27** (Standard generalization in the Bernoulli model.). *Let* $(x, y) \in \mathbb{R}^d \times \{\pm 1\}$ *be drawn from a* $(\theta^\star, \tau)$-*Bernoulli model. Let* $\widehat{w} \in \mathbb{R}^d$ *be the unit vector in the direction of* $z = yx$, *i.e.,* $\widehat{w} = z/\|z\|_2$. *Then with probability at least* $1 - \exp(-\frac{\tau^2 d}{2})$, *the linear classifier* $f_{\widehat{w}}$ *has classification error at most* $\exp(-2\tau^4 d)$.

*Proof.* We invoke Lemma 25 to get

$$\langle \widehat{w}, \theta^\star\rangle \geq \tau\sqrt{d}$$

with probability at least $1 - \exp(-\frac{\tau^2 d}{2})$ as stated in the theorem. Next, unwrapping the definition of $f_{\widehat{w}}$ allows us to write the classification error of $f_{\widehat{w}}$ as

$$\mathbb{P}[f_{\widehat{w}}(x) \neq y] = \mathbb{P}[\langle \widehat{w}, z\rangle \leq 0].$$

Invoking Lemma 26 then gives the desired bound. $\qquad\square$

**Corollary 28** (Generalization from a single sample.). *Let* $(x, y) \in \mathbb{R}^d \times \{\pm 1\}$ *be drawn from a* $(\theta^\star, \tau)$-*Bernoulli model with*

$$\tau \geq \left(\frac{\log 1/\beta}{2d}\right)^{1/4}.$$

*Let* $\widehat{w} \in \mathbb{R}^d$ *be the unit vector* $\widehat{w} = \frac{yx}{\|x\|_2}$. *Then with probability at least* $1 - \exp(-\frac{\tau^2 d}{2})$, *the linear classifier* $f_{\widehat{w}}$ *has classification error at most* $\beta$.

*Proof.* Invoking Theorem 27 gives a classification error bound of

$$\beta' = \exp(-2\tau^4 d).$$

It remains to show that $\beta' \leq \beta$. Now,

$$\log 1/\beta' = 2\tau^4 d$$
$$\geq 2 \cdot \frac{\log 1/\beta}{2d} \cdot d$$
$$\geq \log 1/\beta$$

which yields the desired bound. $\qquad\square$

## F.2 Lower bounds

In this section, we show that any *linear classifier* for the $(\theta^*, \tau)$-Bernoulli model requires many samples to be robust. The main result is formalized in Theorem 31, which can be simplified to yield Theorem 9 from the main text. Before we proceed to the main theorem, we first prove a simple but useful lemma.

**Lemma 29.** *Let* $\theta$ *be drawn uniformly at random from* $\{-1, 1\}$ *and let* $(x_1, y_1), \ldots, (x_n, y_n)$ *be drawn independently from the* $(\theta, \tau)$-*Bernoulli model. Then for* $\tau \leq 1/4$ *and* $n \leq \frac{1}{\tau^2}$, *we have with probability* $1 - \delta$ *over the samples that*

$$\log \frac{\Pr[\theta = +1 \mid (x_1, y_1)\ldots, (x_n, y_n)]}{\Pr[\theta = -1 \mid (x_1, y_1)\ldots, (x_n, y_n)]} \in \left[-15\tau\sqrt{2n\log\frac{2}{\delta}},\ 15\tau\sqrt{2n\log\frac{2}{\delta}}\right]$$

*Proof.* For any sequence $(x_1, y_1) \ldots, (x_n, y_n)$, we can write

$$\frac{\Pr[\theta = +1 \mid (x_1, y_1) \ldots, (x_n, y_n)]}{\Pr[\theta = -1 \mid (x_1, y_1) \ldots, (x_n, y_n)]} = \frac{\Pr[(x_1, y_1), \ldots, (x_n, y_n) \mid \theta = +1]}{\Pr[(x_1, y_1), \ldots, (x_n, y_n) \mid \theta = -1]} \tag{4}$$

because $\Pr[\theta = +1] = \Pr[\theta = -1]$. We now simplify the right hand side to

$$\frac{\Pr[(x_1, y_1), \ldots, (x_n, y_n) \mid \theta = +1]}{\Pr[(x_1, y_1), \ldots, (x_n, y_n) \mid \theta = -1]} = \prod_{i=1}^{n} \frac{\Pr[(x_i, y_i) \mid \theta = +1]}{\Pr[(x_i, y_i) \mid \theta = -1]}$$

$$= \prod_{i=1}^{n} \left( \frac{\frac{1}{2} + \tau}{\frac{1}{2} - \tau} \right)^{y_i x_i} \tag{5}$$

where the second line follows from a simple calculation of the conditional probabilities.

Writing $z_i = y_i x_i$, we next combine Equations (4) and (5) to

$$\frac{\Pr[\theta = +1 \mid (x_1, y_1) \ldots, (x_n, y_n)]}{\Pr[\theta = -1 \mid (x_1, y_1) \ldots, (x_n, y_n)]} = \exp\left( \hat{\tau} \sum_{i=1}^{n} z_i \right),$$

where $\hat{\tau}$ is such that $\exp(\hat{\tau}) = \frac{1+2\tau}{1-2\tau}$. For $\tau \leq \frac{1}{4}$, a simple calculation shows that $\hat{\tau} \leq 5\tau$.

Conditioned on $\theta$, the sum $\overline{z} = \sum_{i=1}^{n} z_i$ has expectation $2\tau n\theta \leq 2\tau n$. Hoeffding's Inequality (e.g., see Theorem 2.8 in [8]) then yields that with probability $1 - \delta/2$

$$\overline{z} \leq 2\tau n + \sqrt{2n \log \frac{2}{\delta}} .$$

It follows that with probability $1 - \delta/2$ (taken over the samples $z_1, \ldots, z_n$), the likelihood ratio above is bounded by

$$\exp(\hat{\tau} \sum_i z_i) \leq \exp\left( 2\hat{\tau}\tau n + \hat{\tau}\sqrt{2n \log \frac{2}{\delta}} \right)$$

Under the assumptions that $n \leq \frac{1}{\tau^2}$, we have

$$\tau n \leq \sqrt{n}$$

and the upper bound follows because the first term in the $\exp$ is at most twice the second term. The lower bound is symmetric. $\square$

We next evaluate the $\ell_\infty$ robustness of the optimal linear classifier.

**Lemma 30.** *Let $\theta^\star \in \{-1, +1\}^d$ and consider the linear classifier $f_{\theta^\star}$ for the $(\theta^\star, \tau)$-Bernoulli model. Then,*

$\ell_\infty^\tau$**-robustness:** *The $\ell_\infty^\tau$-classification error of $f_{\theta^\star}$ is at most $2\exp(-\tau^2 d/2)$.*

$\ell_\infty^{3\tau}$**-nonrobustness:** *The $\ell_\infty^{3\tau}$-classification error of $f_{\theta^\star}$ is at least $1 - 2\exp(-\tau^2 d/2)$.*

**Near-optimality of $\theta^\star$:** *For any linear classifier, the $\ell_\infty^{3\tau}$-classification error is at least $\frac{1}{6}$.*

*Proof.* Let $(x, y)$ be drawn from the $(\theta^\star, \tau)$-Bernoulli model. Then for the linear classifier $w = \theta^\star$, we have

$$\mathbb{E}[\langle w, yx \rangle] = 2\tau \langle w, \theta^\star \rangle = 2\tau d .$$

Let $S$ denote the set

$$S = \{(x, y) : \langle w, yx \rangle \in [\tau d, 3\tau d]\} .$$

Hoeffding's Inequality (e.g., see Theorem 2.8 in [8]) then gives

$$\Pr[(x, y) \notin S] = \Pr[\langle w, yx \rangle \notin [\tau d, 3\tau d]] \leq 2\exp(-\tau^2 d/2) .$$

On the other hand, for a parameter $\varepsilon$,

$$\sup_{e \in B_\infty^\varepsilon} \langle w, e \rangle = \varepsilon \|w\|_1 = \varepsilon d .$$

Thus if $\varepsilon < \tau$, then for any $(x, y) \in S$,

$$\inf_{e \in B_\infty^\varepsilon} \langle w, y(x + e) \rangle > 0 \,,$$

so that any $(x, y) \in S$ is $\ell_\infty^\tau$-robustly classified. On the other hand, for $\varepsilon > 3\tau$, for any $(x, y) \in S$,

$$\inf_{e \in B_\infty^\varepsilon} \langle w, y(x + e) \rangle < 0 \,,$$

so that $(x, y)$ is not $\ell_\infty^{3\tau}$-robustly classified.

Finally, let $w'$ be any other linear classifier. Then we have

$$\mathbb{E}[\langle w', yx \rangle] = 2\tau \langle w', \theta^\star \rangle \leq 2\tau \|w'\|_1 \,,$$

Let $E_i$ be a $\pm 1$ random variable with expectation $2\tau$. We observe that the random variable $yx_i w_i'$ is stochastically dominated by $E_i \cdot |w_i'|$ (note that $yx_i$ is itself a $\pm 1$ random variable with expectation $2\tau$). We can now write $E_i$ as

$$E = A_i + B_i \,,$$

where the random variable $A_i$ is in $\{0, 1\}$ and has expectation $2\tau$. The random variable $B_i$ is in $\{-1, 0, 1\}$ and has a symmetric distribution that depends on $A$. In particular, $B_i = 0$ iff $A_i = 1$ and $B_i$ is a Rademacher random variable otherwise. Since $A_i$ is non-negative, we can use Markov's inequality on $\sum_i |w_i| A_i$. The $B_i$'s have a symmetric distribution even conditioned on $A_i$ so that $\sum_i |w_i'| B_i \leq 0$ with probability at least $1/2$. Thus with probability at least $1/6$, we have

$$\langle w', yx \rangle \leq 3\tau \|w'\|_1 \,.$$

Thus for any $\varepsilon > 3\tau$,

$$\inf_{e \in B_\infty^\varepsilon} \langle w', y(x + e) \rangle = \langle w', yx \rangle + \inf_{e \in B_\infty^\varepsilon} \langle w', ye \rangle$$
$$\leq 3\tau \|w\|_1 - \varepsilon \|w\|_1$$
$$< 0 \,.$$

Thus the $\ell_\infty^{3\tau}$-classification error of $w'$ is at least $1/6$. $\qquad\square$

Lemma 30 implies that the most interesting robustness regime for linear classifiers is $\varepsilon = O(\tau)$. For larger values of $\varepsilon$, it is impossible to learn a linear classifier with small robust classification error regardless of the number of samples used.

We now focus on this robustness regime and establishes a lower bound on the sample complexity of $\ell_\infty^\varepsilon$-robust classification for $\varepsilon \in (0, \tau)$.

**Theorem 31.** *Let $g_n$ be a linear classifier learning algorithm, i.e., a function that takes $n$ samples from $\{-1, +1\}^d \times \pm 1$ to a linear classifier $w \in \mathbb{R}^d$. Suppose that we choose $\theta^\star$ uniformly at random from $\{-1, +1\}^d$ and draw $n$ samples from the $(\theta^\star, \tau)$-Bernoulli model with $\tau \leq 1/4$. Let $w$ then be the output of $g_n$ on these samples. Moreover, let $\varepsilon < 3\tau$ and $0 < \gamma < 1/2$. Then if*

$$n \leq \frac{\varepsilon^2 \gamma^2}{5000 \cdot \tau^4 \log(4d/\gamma)}$$

*the linear classifier $f_w$ has expected $\ell_\infty^\varepsilon$-classification error at least $\frac{1}{2} - \gamma$.*

Before we proceed to the formal proof, we briefly explain the approach at a high level. Informally, Lemma 29 above implies that for small $n$, the algorithm $g_n$ is sufficiently uncertain about each co-ordinate $\theta_i^\star$ so that in expectation, the dot product $\langle w, \theta^\star \rangle$ is small compared to $\|w\|_1$. Since the $\ell_1$ norm $\|w\|_1$ is dual to the $\ell_\infty$ norm bounding the adversarial perturbation, it can be related to the adversarial robustness of the classifier $w$ on a fresh sample $x$. This then leads to the lower bound stated above, as we will now prove in more detail.

*Proof.* Let $S$ be the set of $n$ samples input to $g_n$ and let $w$ be the resulting classifier as defined in the theorem. Our first goal is to bound the uncertainty in the estimate $w$ by establishing an upper bound on $|\mathbb{E}[\theta_i^\star | S]|$ for each $i \in [d]$, which will in turn allow us to bound $\mathbb{E}_{\theta^\star}[\langle w, \theta^\star \rangle | S]$.

We have
$$\mathbb{E}[\theta_i^\star|S] \;=\; \mathbb{P}[\theta_i^\star = +1|S] - \mathbb{P}[\theta_i^\star = -1|S] \,.$$
We first consider the case that $\mathbb{P}[\theta_i^\star = +1|S] \geq \mathbb{P}[\theta_i^\star = -1|S]$, which means that the conditional expectation $\mathbb{E}[\theta_i^\star|S]$ is non-negative. Hence it suffices to provide an upper bound on this quantity. The lower bound in the complementary case $\mathbb{P}[\theta_i^\star = +1|S] < \mathbb{P}[\theta_i^\star = -1|S]$ can be derived analogously.

We have
$$
\begin{aligned}
\mathbb{E}[\theta_i^\star|S] \;&=\; \mathbb{P}[\theta_i^\star = +1|S] - \mathbb{P}[\theta_i^\star = -1|S] \\
&=\; \mathbb{P}[\theta_i^\star = -1|S]\left(\frac{\mathbb{P}[\theta_i^\star = +1|S]}{\mathbb{P}[\theta_i^\star = -1|S]} - 1\right) \\
&\leq\; \frac{1}{2}\left(\frac{\mathbb{P}[\theta_i^\star = +1|S]}{\mathbb{P}[\theta_i^\star = -1|S]} - 1\right)
\end{aligned}
\tag{6}
$$
where we used the assumption $\mathbb{P}[\theta_i^\star = +1|S] \geq \mathbb{P}[\theta_i^\star = -1|S]$ (and hence $\mathbb{P}[\theta_i^\star = -1|S] \leq 1/2$).

Next, we bound the ratio of probabilities by invoking Lemma 29 (note that we have $\tau \leq 1/4$ and $n \leq 1/\tau^2$ as required). With probability $(1 - \frac{\gamma}{2})$, $S$ is such that for all $i \in [d]$ we have

$$\frac{\Pr[\theta_i^\star = +1 \mid S]}{\Pr[\theta_i^\star = -1 \mid S]} \;\in\; \left[\exp\left(-15\tau\sqrt{2n\log\frac{4d}{\gamma}}\right),\; \exp\left(15\tau\sqrt{2n\log\frac{4d}{\gamma}}\right)\right] \,.$$

Substituting this into Equation (6) then yields

$$
\begin{aligned}
\mathbb{E}[\theta_i^\star|S] \;&\leq\; \frac{1}{2}\left(\exp\left(15\tau\sqrt{2n\log\frac{4d}{\gamma}}\right) - 1\right) \\
&\leq\; 15\tau\sqrt{2n\log\frac{4d}{\gamma}}
\end{aligned}
$$

where we used the inequality $e^x - 1 \leq 2x$ for $0 \leq x \leq 1$ (note that the upper bound on $n$ in the theorem implies that the argument to the exponential function is in this range).

Combining the bound above with the analogous lower bound gives

$$|\mathbb{E}[\theta_i^\star|S]| \;\leq\; 15\tau\sqrt{2n\log\frac{4d}{\gamma}}$$

so that

$$
\begin{aligned}
\mathbb{E}_{\theta^\star}[\langle w, \theta^\star\rangle|S] \;&=\; \sum_{i=1}^{d}\mathbb{E}_{\theta^\star}[w_i\theta_i^\star|S] \\
&=\; \sum_{i=1}^{d} w_i \cdot \mathbb{E}_{\theta^\star}[\theta_i^\star|S] \\
&\leq\; 15\tau\sqrt{2n\log\frac{4d}{\gamma}} \cdot \|w\|_1 \,.
\end{aligned}
$$

We condition on such an $S$ for the rest of this proof.

The second part of the proof will bound the classification margin the linear classifier $w$ achieves on a fresh sample $x$. Incorporating the class label $y$, this margin is the quantity $y\langle w, x\rangle$. From the first part of the proof, it follows that

$$
\begin{aligned}
\mathbb{E}_{\theta^\star,(x,y)}[\langle w, yx\rangle] \;&=\; 2\tau \cdot \mathbb{E}_{\theta^\star}[\langle w, \theta^\star\rangle] \\
&\leq\; 30\tau^2\sqrt{2n\log(4d/\gamma)} \cdot \|w\|_1 \,.
\end{aligned}
$$

To simplify the following calculation, we introduce the shorthand $a_n = 30\tau^2\sqrt{2n\log(4d/\gamma)}$. Next, we provide a tail bound on $\langle w, yx\rangle$. Similar to Lemma 30, we observe that the random variable $yx_i w_i$

is stochastically dominated by $E_i \cdot |w_i|$ where $E_i$ is a $\pm 1$ random variable with expectation $a_n$. We can again write $E_i$ as

$$E = A_i + B_i ,$$

where the random variable $A_i$ is in $\{0,1\}$ and has expectation $a_n$. The random variable $B_i$ is in $\{-1,0,1\}$ and has a symmetric distribution that depends on $A$. In particular, $B_i = 0$ iff $A_i = 1$ and $B_i$ is a Rademacher random variable otherwise. Since $A_i$ is non-negative, we can use Markov's inequality on $\sum_i |w_i| A_i$. The $B_i$'s have a symmetric distribution even conditioned on $A_i$ so that $\sum_i |w_i| B_i \leq 0$ with probability at least $^1/_2$. Thus with probability at least $\frac{1-\gamma}{2}$, we have

$$\langle w, yx \rangle \leq \frac{a_n}{\gamma} \|w\|_1 .$$

Using the upper bound on $n$ from the theorem statement, we have

$$\frac{a_n}{\gamma} \leq \frac{30\tau^2 \sqrt{2n \log(4d/\gamma)}}{\gamma}$$

$$< \varepsilon .$$

Next, consider the strongest adversarial perturbation $e \in \mathcal{B}_\infty^\varepsilon$ for a given $w$, i.e., the vector $e \in \mathbb{R}^d$ achieving

$$\min_{e \in \mathcal{B}_\infty^\varepsilon} \langle w, e \rangle .$$

By duality, the minimum value is exactly $\varepsilon \|w\|_1$. Hence conditioned on the samples $S$ and the bound on $\langle w, yx \rangle$, the adversarially perturbed point $x + e$ is mis-classified because

$$y\langle w, x \rangle = y\langle w, x \rangle + y\langle w, e \rangle$$

$$< \varepsilon \|w\|_1 - \varepsilon \|w\|_1$$

$$= 0 .$$

The overall probability of this event occuring is at least $1 - \frac{\gamma}{2}$ (conditioning on $S$) times $\frac{1-\gamma}{2}$ (bound on $\langle w, x \rangle$). Since

$$\left(1 - \frac{\gamma}{2}\right)\left(\frac{1-\gamma}{2}\right) \geq \frac{1}{2} - \gamma .$$

this completes the proof. $\qquad\qquad\qquad\qquad\qquad\qquad\qquad\qquad\qquad\qquad\qquad\qquad\square$

## G    Additional Experiment Details

**Model Architecture**    For MNIST, we use the simple convolution architecture obtained from the TensorFlow tutorial [1]. In order to prevent the model from overfitting when trained on small data samples, we regularize the model by adding weight decay with parameter $0.5$ to the training loss. For CIFAR-10, we consider a standard ResNet model [23]. It has 4 groups of residual layers with filter sizes $(16, 16, 32, 64)$ and 5 residual units each. On SVHN, we also trained a network of larger capacity (filter sizes of $(16, 64, 128, 256)$ instead of $(16, 16, 32, 64)$) in order to perform well on the harder problems with larger adversarial perturbations. All of our models achieve close to state-of-the-art performance on the respective benchmark.

**Robust optimization.**    We perform robust optimization to train our classifiers. In particular, we train against a projected gradient descent (PGD) adversary, starting from a random initial perturbation of the training datapoint (see [36] for more details). We consider adversarial perturbations in $\ell_\infty$ norm, performing PGD updates of the form

$$x_{t+1} = \Pi_{\mathcal{B}_\infty^\varepsilon(x_0)} \left(x_t + \lambda \cdot \mathrm{sgn}(\nabla \mathcal{L}(x_t))\right)$$

for some step size $\lambda$. Here, $\mathcal{L}$ denotes the loss of the model, while $\Pi_{\mathcal{B}_\infty^r(x)}(z)$ corresponds to projecting $z$ onto the $\ell_\infty$ ball of radius $r$ around $x$. On MNIST, we perform 20 steps of PGD, while on CIFAR-10 and SVHN we perform 10 steps. We evaluate all networks against a 20-step PGD adversary. We choose the PGD step size to be $2.5 \cdot \varepsilon/k$, where $\varepsilon$ denotes the maximal allowed perturbation and $k$ is the total number of steps. This allows PGD to reach the boundary of the optimization region within $\frac{k}{2.5}$ steps from any starting point.

## H    Omitted Figures

Figure 4: Complete experiments for adversarially robust generalization for $\ell_\infty$ adversaries. For each dataset and training $\varepsilon$ we report the performance of the corresponding classifier for each testing $\varepsilon$. We observe that the best performance on natural examples is achieved through natural training and the best adversarial performance is achieved when training with the largest $\varepsilon_{train}$ considered.

Figure 5: Complete experiments for adversarially robust generalization for $\ell_\infty$ adversaries for standard networks (*top row*) and networks with thresholding (*bottom row*) for the MNIST dataset. Thresholding corresponds to replacing the first convolutional layer with two channels ReLU$(x - \varepsilon)$ computing ReLU$(x - (1 - \varepsilon))$. For each training $\varepsilon_{train}$ we report the performance of the corresponding classifier for each testing $\varepsilon_{test}$. For natural training, we use thresholding filters identical to those used for $\varepsilon_{train} = 0.1$. We observe that in each case, explicitly encoding thresholding filters in the network architecture boosts the adversarial robustness for a given training $\varepsilon_{train}$ and training set size.

## Footnotes

[5]It is worth noting that the distribution in [20] has only one degree of freedom. Hence we conjecture that the observed difficulty of robust learning in their setup is due to the chosen model class and not due to an information-theoretic limit as in our work.

[6]For a set $A$ and a vector $v$, we use the notation $A+v$ to denote the set $\{x+v:x\in A\}$.