[Reviews · NeurIPS 2018]

Reviewer 1



The paper considered theoretical results on adversarially robust generalization, which studies the robustness of classifiers in the presence of even small noise. In particular, the work studied the generalization of adversarially robust learning by investigating the sample complexity in a comparison to that of standard learning. Specifically, the study focused on two simple concrete distribution models: gaussian model and Bernoulli model. For both models, the authors established the lower and upper bounds for the sample complexities. From these results, they drew the conclusion that the sample complexity of robust generalization is much larger than standard generalization. The results are interesting and potentially of great interests to machine learning community. The proof techniques are non-trivial involving delicate properties of Gaussian, Bernoulli distributed random variables. One possible weakness of the paper is that the results are established based on very special models. Other comments: 1. In line 178-179, the sample complexity of robust generalization is claimed to be larger than of standard generalization by the square root of d. Could you explain this more on this claim? I would think that one should compare the sample complexities assuming the classification errors are at the same level -- the results of Theorem 5 and 6 certainly used different classification errors to derive the sample complexity bounds. The same comment also applies to the claim stated in lines 229-231. 2. I am not sure if these models are practically realistic. Usually, one will start from the marginal distribution to sample inputs x and then the conditional distribution P(y|x) to sample the outputs y given x. The Gaussian and Bernoulli models (definitions 1 and 2) used the other way around -- y is randomly drawn and then x is from Gaussian or Bernoulli. Any responses from the authors will be appreciated. I went through the feedback and I am satisfied with their response, in particular on the special models. I recommend its acceptance.

Reviewer 2



The paper analyses the robustness of machine learning algorithms to adversarial examples. For two particular artificial data distributions of binary classification data---overlapping Gaussians and overlapping Bernoulli distributions---the paper analyzes the sample complexity required to achieve robustness against adversarial examples. This sample complexity is compared to classical ones for the generalization error. The theoretical findings are supported by experiments on datasets that share properties with the artificial data distributions. The quality of the paper is good, the theoretical parts are sound and well-explained in the appendix. Even though the analyzed data distributions are very limited, the results are still very insightful; in particular, because they hold for arbitrary learning algorithms. The empirical evaluation is insightful and properly related to the theoretical findings. The paper is well-written and clear. The theoretical results are undertandable, their proofs are comprehensible. I have some minor comments to improve the clarity at the end of this review. The main contribution of the paper are robust sample complexity bounds for two specific data distributions for binary classification for arbitrary learning algorithms (and model classes). This adds considerably to the understanding of robustness against adversarial examples. The empirical evalutation support these theoretical results. In contrast to the other theoretical findings the results on thresholding appear to be a bit trivial though: if you map every perturbation of a binary vector back to a binary vector using thresholding, then trivially the perturbations have no effect as long as they do not change the sign of elements of the binary vector. The paper is a useful contribution to the community, because it extends the understanding of robustness against adversarial examples. Since the theoretical analysis holds for arbitrary learning algorithms, it emphasizes that adversarial examples are a general problem for learning algorithms that can be tackled by using more training examples. However, since max-margin classifiers are known to be more robust against adversarial examples, it would be great to discuss them in relation to especially deep neural networks in more detail. The author's response has addressed my concerns about the thresholding approach, as well as my remark to further investigate max-margin classifiers. Detailed Comments: - The sample complexity of robust generalization is larger by sqrt(d), i.e., sub-linear in the dimension, not polynomial. - I would have liked the authors to explain more on the relation of the sample complexity of max-margin models to the robust sample complexity. - In theorem 4, I think \widehat{w} = yx should be \widehat{w} = yx / \|x\|_2, same in Theorem 5. - Line 160: the reference Corollary 23 should be Corollary 22. - In Theorem 4 and 8 it should be stated that achieving this bound with a single example is only possible because of the symmetrie of the positive and negative class distributions (at first glance, the result was a bit surprising). - I can see some benefit in shortening the formulas in Theorem 4 and Theorem 5 using constants c, c_1, c_2. However, since c_1 = 1/32 and c_2 = 64, they could also be included. Same for c= (5\sqrt{\log 1/\beta})^-1 for \beta = 0.01 (i.e., c = 5\sqrt(2)), where even the desired error bound \beta is part of the constant. - Some restrictions on possible variable assignments are not given explicitely: e.g., in Corollary 19, the error is assumed to be < 1, in Corollary 22 the \epsilon-Radius is assumed to be <= 1/4. - In the introduction, when introducing l_\infty perturbations, I suggest noting that this means robustness against adversarial examples generated within the l_\infty hypercube around a given example. - In the motivating example in the introduction: since Figure 1 shows accuracies, I suggest not using the term error---even though the two are related---since it is a bit confusing at first. Also, it might be interesting to also see the standard train accuracy.

Reviewer 3



This paper considers the sample complexity to learn robust classifiers. It studies the problem in the l_\infty attack setting and in two data models, the first assuming the data is from a mixture of two class-conditional Gaussians, and the second assuming the data is from some distributional model with binary features. In the first model, it gives sample complexity bounds for standard classification error and for robust classification error. Most interestingly, it provides an information-theoretic lower bound showing that a much larger number of examples are necessary for learning robust classifiers. The blowup factor can be as large as \sqrt{d} where d is the dimension of the data. In the second model, it shows that if linear classifiers are used then a similar increase in sample complexity is observed. But with a variant with thresholding, the sample complexity for robust error can be similar to that of the standard error. The work also provides a range of experiments on several practical datasets frequently used in adversarial ML. Pros: 1. The topic is interesting and important. As far as I know, it has not been systematically studied before (from the theoretical or empirical side). The work provides interesting results along this direction. 2. The lower bound in the Gaussians model is very interesting. It's information-theoretic, so it suggests that the vulnerability to attacks can stem from the sample size, not necessarily from algorithmic design. It's proved in a very simple model so it's reasonable that this insight carries over to more general/practical cases. It's also practically useful, suggesting that one needs to check if the dataset size is large enough for robustness, before trying to improve the learning algorithms. Cons: I'm not fully convinced that the analysis in the second model explains why adversarial robustness can be achieved on MNIST with low sample complexity. First, the model assumes that conditioned on the class, the pixels are independent. This abstracts away the semantic relations between different pixels in the image, which intuitively should be an important factor for achieving robustness. Second, there are empirical results on datasets CIFAR10 and ImageNet showing that discretizing the pixels to a few fixed discrete values does not hurt the standard error, but does not help improve the robustness. It suggests that only discretization/thresholding may not be sufficient.